# Precise asymptotics of reweighted least-squares algorithms for linear diagonal networks

**Chiraag Kaushik**
Electrical and Computer Engineering
Georgia Institute of Technology
Atlanta, GA 30308
ckaushik7@gatech.edu

**Justin Romberg**
Electrical and Computer Engineering
Georgia Institute of Technology
Atlanta, GA 30308
jrom@ece.gatech.edu

**Vidya Muthukumar**
Electrical and Computer Engineering,
Industrial & Systems Engineering
Atlanta, GA 30308
vmuthukumar8@gatech.edu

## Abstract

The classical *iteratively reweighted least-squares* (IRLS) algorithm aims to recover an unknown signal from linear measurements by performing a sequence of weighted least squares problems, where the weights are recursively updated at each step. Varieties of this algorithm have been shown to achieve favorable empirical performance and theoretical guarantees for sparse recovery and $\ell_p$-norm minimization. Recently, some preliminary connections have also been made between IRLS and certain types of non-convex linear neural network architectures that are observed to exploit low-dimensional structure in high-dimensional linear models. In this work, we provide a unified asymptotic analysis for a family of algorithms that encompasses IRLS, the recently proposed lin-RFM algorithm (which was motivated by feature learning in neural networks), and the alternating minimization algorithm on linear diagonal neural networks. Our analysis operates in a "batched" setting with i.i.d. Gaussian covariates and shows that, with appropriately chosen reweighting policy, the algorithm can achieve favorable performance in only a handful of iterations. We also extend our results to the case of group-sparse recovery and show that leveraging this structure in the reweighting scheme provably improves test error compared to coordinate-wise reweighting.

## 1 Introduction

Many high-dimensional machine learning and signal processing tasks rely on solving optimization problems with regularizers that explicitly enforce certain structure on the learned parameters. The traditional formulation for such tasks involves a regularized empirical risk minimization (ERM) problem of the form

$$\min_{\boldsymbol{\theta} \in \mathbb{R}^d} L(\boldsymbol{\theta}) + \lambda R(\boldsymbol{\theta}), \tag{1}$$

where $L(\cdot)$ is a loss function that encourages fidelity to the observed training data and $R(\cdot)$ encodes desirable structural properties. In many important applications, it is desirable to obtain a sparsity-seeking solution; in such cases, the regularizer is typically non-smooth, as in the LASSO, group LASSO, and nuclear norm regularizers. As an alternative approach to this non-smooth optimization, several recent works have proposed the "Hadamard over-parameterization" of $\boldsymbol{\theta}$ into the entry-wise product of two factors $\boldsymbol{u} \odot \boldsymbol{v}$. While the resulting minimization problem is non-convex, this

parameterization, coupled with a *smooth* regularizer, has been shown to achieve competitive empirical performance (in terms of numerical stability, robustness, and convergence rate) when compared to traditional sparse recovery algorithms [13, 25]. For example, rather than solving the convex, but non-smooth LASSO (where $L$ is the squared loss and $R$ is the $\ell_1$ norm), the Hadamard reparameterization yields the following non-convex and smooth formulation:

$$\min_{\boldsymbol{u},\boldsymbol{v}\in\mathbb{R}^d} L(\boldsymbol{u}\odot\boldsymbol{v}) + \frac{\lambda}{2}(\|\boldsymbol{u}\|_2^2 + \|\boldsymbol{v}\|_2^2). \tag{2}$$

In the case where the regression function is linear in $\boldsymbol{\theta}$, solving (2) is equivalent to learning a function of the form

$$f_{\boldsymbol{u},\boldsymbol{v}}(\boldsymbol{x}) = \langle \boldsymbol{x}, \boldsymbol{u}\odot\boldsymbol{v}\rangle = \langle \mathrm{diag}(\boldsymbol{v})\boldsymbol{x}, \boldsymbol{u}\rangle,$$

which can be thought of as a one hidden layer neural network with linear activation function and inner weight matrix $\mathrm{diag}(\boldsymbol{v})$. In this context, this *linear diagonal neural network (LDNN)* architecture has also been studied as an illustrative case study to improve our understanding of how neural networks perform iterative *"feature learning"* to leverage low-dimensional structure in high-dimensional settings [34, 23]. We note here that, in the linear model case, feature learning is equivalent to learning which subset of the input's coordinates are relevant for the true predictor (i.e., feature selection).

One way to understand the connection between classical sparse recovery algorithms and the Hadamard product/LDNN form in (2) is to consider the change of variable $v_i \to \sqrt{\eta_i}$ and $u_i \to \frac{\theta_i}{\sqrt{\eta_i}}$ [25]. This yields the following optimization problem, which is jointly convex in $\boldsymbol{\eta}$ and $\boldsymbol{\theta}$:

$$\min_{\boldsymbol{\theta}\in\mathbb{R}^d} \min_{\boldsymbol{\eta}\in\mathbb{R}_+^d} L(\boldsymbol{\theta}) + \frac{\lambda}{2}\sum_{i=1}^{d}\left(\frac{\theta_i^2}{\eta_i} + \eta_i\right). \tag{3}$$

After solving the minimum over $\boldsymbol{\eta}$ explicitly, the second term becomes exactly $\lambda\|\boldsymbol{\theta}\|_1$, and we recover the Lasso objective. This is a special case of the so-called "eta-trick" [1], which can be used to write many common sparsity-inducing penalties as the minimization of a quadratic functional of $\boldsymbol{\theta}$.

A variety of algorithms for learning Hadamard product parameterizations have recently been studied, including alternating minimization [13], bi-level optimization [25], and joint gradient descent on $(\boldsymbol{u},\boldsymbol{v})$ [34]. The connection to the $(\boldsymbol{\theta},\boldsymbol{\eta})$ optimization in (3) can also be leveraged to construct algorithms based on classical sparse recovery techniques. In particular, alternating minimization over $\boldsymbol{\theta}$ and $\boldsymbol{\eta}$ in (3) yields the popular *iteratively reweighted least-squares (IRLS)* algorithm [11, 9]. Translating these updates to the equivalent updates on $(\boldsymbol{u},\boldsymbol{v})$, we obtain an iterative least-squares algorithm for LDNNs, which alternately sets $\boldsymbol{v}^{(t+1)} = \sqrt{|\boldsymbol{u}^{(t)}\odot\boldsymbol{v}^{(t)}|}$ and performs a weighted least squares update on $\boldsymbol{u}$. This particular form of reparameterized IRLS was generalized in [28] to a larger family of iterative least-squares algorithms under the name of *linear recursive feature machines (lin-RFM)*.

While several methods for learning Hadamard/LDNN parameterizations have been introduced in the literature, there remain many open questions about how they each perform and how they compare. Theoretical analyses of these algorithms typically assume fixed, possibly worst-case training data, and aim to characterize the properties of the fixed-points [28, 34] or give convergence guarantees to second-order stationary points [25]. However, these worst-case analyses do not readily yield guarantees on the estimation error, which is the principal metric of interest. Indeed, many works have shown that studying the *average-case*, or typical, behavior of non-convex optimization algorithms can allow for estimation guarantees that are more precise and reflective of practice [14, 18, 7].

In this paper, we provide a precise analysis of a general family of iterative algorithms for learning LDNNs that take the form

$$\boldsymbol{u}^{(t+1)} = \arg\min_{\boldsymbol{u}\in\mathbb{R}^d} \frac{1}{n}\left\|\boldsymbol{y}^{(t)} - \frac{1}{\sqrt{d}}\boldsymbol{X}^{(t)}(\boldsymbol{u}\odot\boldsymbol{v}^{(t)})\right\|_2^2 + \frac{\lambda}{d}\|\boldsymbol{u}\|_2^2$$

$$\boldsymbol{v}^{(t+1)} = \psi(\boldsymbol{u}^{(t+1)}, \boldsymbol{v}^{(t)}),$$

for some reweighting function $\psi$ and batches of training data $(\boldsymbol{X}^{(t)}, \boldsymbol{y}^{(t)})$. As we show in Section 2, this formulation encompasses multiple existing algorithms, including reparameterized IRLS, lin-RFM, and alternating minimization over $\boldsymbol{u}$ and $\boldsymbol{v}$. We consider the common scenario where training is performed with batches of data and characterize the *exact* distribution of the parameters after each iteration in the high-dimensional limit $(n, d) \to \infty$. This allows us to address questions such as

- How do different algorithm choices compare (in terms of convergence and signal recovery) in the high-dimensional regime?

- How many iterations does it take common algorithms to find statistically favorable solutions?

- What is the effect of *model architecture* in LDNNs? Does leveraging group structure provably improve sample complexity when the ground-truth signal is group-sparse?

**Contributions:** We define a general class of algorithms which learns LDNNs by alternately performing least-squares and reweighting steps in a sample-split/batched setting, and we show the following.

**(1)** Under mild assumptions on the target signal, initialization, and reweighting function, we provide an exact characterization of the distribution of the entries of the parameters at each iteration in the limit as $n, d$ approach infinity (Theorem 1).

**(2)** We show that this asymptotic result aligns well with numerical simulations and allows for accurate prediction of the test error at each iteration. This enables rigorous comparison between different algorithms and demonstrates that, with appropriate reweighting schemes, a statistically favorable solution can be obtained in only a handful of iterations.

**(3)** Lastly, we extend our asymptotic framework to a setting of *structured sparsity*, where $\theta^*$ has group-sparse structure (Theorem 2). Our results show that using a grouped Hadamard parameterization (i.e., tying together groups of weights in the LDNN) effectively learns such signals, with performance scaling with the number of non-zero groups, rather than the total sparsity level.

## 1.1 Related work

**IRLS and the $\eta$-trick:** The reformulation of non-smooth regularizers in terms of quadratic variational forms (the "$\eta$-trick") has been studied in various early works in computer vision and robust statistics [12, 4]. Further analysis and examples of sparsity-promoting norms are provided in [20, 2], and [25] provides a characterization of when a regularizer admits a variational form of this type. The resultant optimization algorithm is iteratively-reweighted least-squares (IRLS), a popular technique for compressive sensing and sparse recovery [11, 9]. These works also consider IRLS algorithms corresponding to $\ell_p$-norm regularization for $0 < p < 1$; in this case, the minimization is no longer convex, but [11] shows that such methods can find sparse solutions with fast local convergence rate. The family of algorithms we consider includes a reparameterized version of each of these IRLS algorithms, but unlike these prior works, we consider a batched setting and the high-dimensional asymptotic regime. Moreover, our results apply to other algorithms which may not be easily expressed as resulting from the $\eta$-trick.

**Hadamard parameterization and linear diagonal networks:** The reparameterization of $\theta$ into the product of factors $u \odot v$ has been considered in a variety of recent works. The authors of [31, 35] show that early-stopped joint gradient descent over the two factors can lead to optimal sample complexity for sparse linear regression. The equivalence of this parameterization to LDNNs has also led to a surge of interest in the *implicit bias* of gradient descent/flow on this parameterization, i.e., a characterization of which solution gradient descent will reach without explicit regularization (corresponding to $\lambda = 0$). These works typically consider gradient flow run until completion and characterize the solution as a minimizer of a certain sparsity-inducing functional that depends on the initialization [34, 10, 23].

The connection between the LASSO (as well as some non-convex $\ell_q$ penalties) and the Hadamard parameterization was studied in [13], where alternating minimization over the two factors is used instead of first-order methods. More recently, [25] extends these observations by making explicit the connection to the $\eta$-trick and showing that saddle points are strict (escapable). These insights lead to global convergence guarantees and a smooth bi-level optimization scheme [25, 26] for non-smooth structured optimization problems that was shown to perform competitively with state-of-the-art solvers. The non-convex landscape of such formulations is further explored in [15], where it is shown that for a large class of parameterizations (including grouped, deep, and fractional Hadamard products), the non-convex problem has no spurious local minima. Motivated by the type of feature learning observed in neural networks, the authors of [28] propose lin-RFM, which updates one of the parameters via weighted least-squares while iteratively updating the other parameter via a reweighting scheme based on the average gradient outer product of the learned function. The authors characterize

properties of the fixed-points and show that, for certain reweighting schemes, lin-RFM is equivalent to a reparameterization of IRLS. The family of algorithms we consider is similar, consisting of a weighted least-squares step and a reweighting step; however, it is more general and doesn't require the reweighting function to have the particular form required by lin-RFM. Moreover, our asymptotic characterization of the iterates allows for a precise understanding of how the test error evolves. On the other hand, our analysis relies on batching/sample-splitting of training data while all of the above works reuse the entire batch of training data at each iteration.

We make particular note here of the few works which explicitly consider a "grouped" Hadamard parameterization, which we consider in Section 4. This corresponds to a LDNN with groups of tied weights in the hidden layer. Early stopped gradient flow/descent for this type of architecture was shown in [17] to achieve sample-complexity scaling with the number of non-zero groups (rather than the overall sparsity). The non-convex landscape for this grouped architecture is studied in [36] and [15]. Our results complement these works by studying group-reweighted least-squares algorithms (rather than gradient methods) for learning functions of this form.

**Precise characterization of higher-order non-convex optimization problems:** On a technical level, our work provides a precise deterministic characterization of a family of higher-order optimization algorithms. In this sense, our results are of a similar flavor to [7], where Gaussian comparison inequalities are used to obtain a precise characterization of non-convex optimization problems. However, since the Hadamard parameterization is a re-parameterization of the actual estimator of interest ($\boldsymbol{\theta} := \boldsymbol{u} \odot \boldsymbol{v}$), the results of [7] are not directly applicable. While our results are asymptotic and do not provide finite-sample guarantees, we provide a *distributional* characterization of $\boldsymbol{v}$ after each reweighting step, which allows us to characterize the behavior of more complicated functions of the iterates. Precise characterizations of alternating minimization and lin-prox methods for rank-1 matrix sensing are studied in the works [6, 19]. While these works obtain non-asymptotic guarantees, the estimation model and resulting optimization objective are quite different, with each unknown parameter interacting with independent sensing vectors (rather than a single sensing vector interacting with the product of the two parameters).

## 2 Background and formulation

**Notation:** The ones vector of dimension $d$ is denoted as $\boldsymbol{1}_d$. We denote the element-wise multiplication (Hadamard product) of two vectors $\boldsymbol{x}$ and $\boldsymbol{y}$ as $\boldsymbol{x} \odot \boldsymbol{y}$. Element-wise division of two vectors is denoted as $\frac{\boldsymbol{x}}{\boldsymbol{y}}$. We say a function $f \colon \mathbb{R}^p \to \mathbb{R}$ is *pseudo-Lipschitz* of order 2 if, for all $\boldsymbol{x}, \boldsymbol{y} \in \mathbb{R}^p$,

$$|f(\boldsymbol{x}) - f(\boldsymbol{y})| \le C(1 + \|\boldsymbol{x}\|_2 + \|\boldsymbol{y}\|_2)\|\boldsymbol{x} - \boldsymbol{y}\|_2$$

for some constant $C > 0$. The set of such functions is denoted by PL(2).

Convergence in probability of a sequence of random variables $X_d$ to a random variable $X$ is denoted by $X_d \overset{P}{\to} X$. Convergence in Wasserstein-2 distance of a sequence of probability distributions $\nu_d$ to a limiting distribution $\nu$ is denoted as $\nu_d \overset{\mathcal{W}_2}{\to} \nu$, and this fact is equivalent to the statement $\mathbb{E}_{X \sim \nu_d} g(X) \to \mathbb{E}_{X \sim \nu} g(X)$ for all $g \in \mathrm{PL}(2)$ [3]. If the $\nu_d$ are *random* probability measures, we say that $\nu_d \overset{\mathcal{W}_2}{\to} \nu$ if the same convergence holds in probability, i.e., $\mathbb{E}_{X \sim \nu_d} g(X) \overset{P}{\to} \mathbb{E}_{X \sim \nu} g(X)$ for all $g \in \mathrm{PL}(2)$. The empirical distribution of a vector $\boldsymbol{z} \in \mathbb{R}^d$ is defined as $\frac{1}{d} \sum_{i=1}^d \delta(z_i)$, where $\delta(z_i)$ is the Dirac delta distribution centered at $z_i$.

**Formulation:** We consider a batched noisy linear model where, at each time $t = 0, 1, \ldots, T$, a user has access to an independent batch of data $(\boldsymbol{X}^{(t)}, \boldsymbol{y}^{(t)}) \in \mathbb{R}^{n \times d} \times \mathbb{R}^n$ satisfying

$$\boldsymbol{y}^{(t)} = \frac{1}{\sqrt{d}} \boldsymbol{X}^{(t)} \boldsymbol{\theta}^* + \boldsymbol{\epsilon}^{(t)}.$$

Above, $\boldsymbol{\theta}^* \in \mathbb{R}^d$ is an unknown signal, $\boldsymbol{X}^{(t)}$ has i.i.d. standard Gaussian entries, and $\boldsymbol{\epsilon}^{(t)} \sim \mathcal{N}(\boldsymbol{0}, \sigma^2 \boldsymbol{I}_n)$ is i.i.d. noise in the measurements. Given an initial weight vector $\boldsymbol{v}^{(0)} \in \mathbb{R}^d$, we are interested in the behavior of iterative algorithms of the form

$$
\begin{aligned}
\boldsymbol{u}^{(t+1)} &= \arg\min_{\boldsymbol{u} \in \mathbb{R}^d} \frac{1}{n} \left\| \boldsymbol{y}^{(t)} - \frac{1}{\sqrt{d}} \boldsymbol{X}^{(t)}(\boldsymbol{u} \odot \boldsymbol{v}^{(t)}) \right\|_2^2 + \frac{\lambda}{d} \|\boldsymbol{u}\|_2^2 \\
\boldsymbol{v}^{(t+1)} &= \psi(\boldsymbol{u}^{(t+1)}, \boldsymbol{v}^{(t)}),
\end{aligned}
\tag{4}
$$

Table 1: Some algorithms taking the form (4)

| Algorithm | Reweighting function |
|-----------|---------------------|
| Alternating minimization (AM) [13] | $\psi(u,v) = u$ |
| Reparameterized IRLS [9, 11, 28] | $\psi(u,v) = (u^2v^2 + \epsilon)^\alpha$ |
| Linear recursive feature machines (lin-RFM) [28] | $\psi(u,v) = \phi(u^2v^2)$ |

where $\psi \colon \mathbb{R} \times \mathbb{R} \to \mathbb{R}$ is a "reweighting" function that acts entry-wise on $(\boldsymbol{u}^{(t)}, \boldsymbol{v}^{(t)})$ and $\lambda > 0$ is a hyperparameter governing the strength of the regularization. We we will study the behavior of the iterates $\boldsymbol{u}^{(t)}, \boldsymbol{v}^{(t)}$ in the high-dimensional limit where $n$ and $d$ both approach infinity with fixed ratio $\frac{d}{n} = \kappa$. Since our primary interest is to reveal the feature learning capabilities of such algorithms when $\boldsymbol{\theta}^*$ is a high-dimensional signal with low-dimensional structure, we will typically focus on the regime where $\kappa > T$, where $T$ is the number of total iterations. This ensures that the total number of observed samples $nT$ is smaller than the ambient dimension $d$.

Before proceeding to our main results, we note that this formulation encompasses a wide variety of classical and modern algorithms (summarized in Table 1):

- **Alternating minimization:** One perspective on this algorithm is to consider it as a way to perform alternating minimization on the non-convex loss function

$$L(\boldsymbol{u}, \boldsymbol{v}) = \frac{1}{n}\left\|\boldsymbol{y} - \frac{1}{\sqrt{d}}\boldsymbol{X}(\boldsymbol{u} \odot \boldsymbol{v})\right\|_2^2 + \frac{\lambda}{d}\|\boldsymbol{u}\|_2^2 + \frac{\lambda}{d}\|\boldsymbol{v}\|_2^2.$$

  Using the fact that the loss function is symmetric in $\boldsymbol{u}$ and $\boldsymbol{v}$, choosing $\psi(u,v) = u$ recovers the mini-batched alternating minimization algorithm for this loss. In other words, $\psi$ simply switches the two parameters $\boldsymbol{u}$ and $\boldsymbol{v}$.

- **IRLS algorithms for sparse recovery:** As shown in [28], classical IRLS reweighting schemes used for sparse recovery and compressed sensing [11, 21] can be reparameterized in the form of (4) $\psi(u,v) = (u^2v^2 + \epsilon)^\alpha$, where different choices of $\alpha$ correspond to different $\ell_p$ penalties. In particular, the choice $p = 2 - 4\alpha$ corresponds to the IRLS-p algorithm of [21].

- **Lin-RFM [28]:** Generalizing the reparameterized IRLS update, the authors of [28] propose the choice $\psi(u,v) = \phi(u^2v^2)$ for some continuous function $\phi \colon \mathbb{R} \to \mathbb{R}^+$. Here, the quantity $u^2v^2$ arises from the average outer product of the gradient of the learned regression function, which was shown empirically in [27] to correlate with the features learned in the weight matrices of various common neural network architectures.

Our goal is to understand statistical properties of the iterates for different choices of $\psi$, and in particular how the test error evolves from iteration to iteration. In the following section, we develop an asymptotic characterization of the iterates that can be used to gain insight into these questions for a large class of reweighting functions and problem settings.

## 3 A precise characterization of the iterates

In this section, we provide a precise characterization of the iterates of the algorithm in Equation 4 with i.i.d. Gaussian covariates. First, we introduce and discuss the two main assumptions needed for our main result. The first assumption is concerned with the distribution of the initialization $\boldsymbol{v}_0$ and the target signal $\boldsymbol{\theta}^*$:

**Assumption 1.** *The empirical distribution of the entries of $\boldsymbol{v}^{(0)}$ and $\boldsymbol{\theta}^*$ converges in $\mathcal{W}_2$ distance to some joint distribution $\Pi_0$, i.e., $\frac{1}{d}\sum_{i=1}^d \delta(v_i^{(0)}, \theta_i^*) \overset{\mathcal{W}_2}{\to} \Pi_0$. Additionally, $v_i^{(0)} \neq 0$ for all $i$ and $\boldsymbol{\theta}^*$ has bounded entries almost surely.*

Here, the requirement of empirical distribution convergence is easily satisfied by common choices of $\boldsymbol{v}^{(0)}$, including the ones vector and i.i.d. Gaussian entries. For a typical sparse regression setup, we might, for example, consider the $\Pi_0$ induced by choosing $\boldsymbol{v}^{(0)} = \mathbf{1}_d$ and letting $\boldsymbol{\theta}^*$ have i.i.d. entries

that equal 0 with certain probability. The requirement that $\boldsymbol{\theta}^*$ has bounded entries appears to be an artifact of the proof, and is used only in the proof of one technical lemma. In our simulations, we find that our asymptotic predictions often remain accurate when $\boldsymbol{\theta}^*$ has entries from distributions which are not bounded almost surely (e.g., Gaussian entries).

Secondly, we define the set of reweighting functions $\psi$ for which our result will apply.

**Assumption 2.** *The reweighting function $\psi\colon \mathbb{R} \times \mathbb{R} \to \mathbb{R}$ satisfies the following:*

1. *If $U, V$ are random variables such that $U, V \neq 0$ with probability 1, then $\psi(U, V) \neq 0$ with probability 1.*

2. *$\psi$ is continuous and bounded or $\psi^2$ is pseudo-Lipschitz of order 2.*

This family allows us to consider many of the choices of $\psi$ discussed in the previous section, including $\psi(u, v) = u$ (AM on linear diagonal networks), $\psi(u, v) = \sqrt{|uv|}$ (lin-RFM and IRLS), $\psi(u, v) = \phi(u^2 v^2)$ for bounded $\phi$ (lin-RFM). We note that this does *not* include some choices of $\psi$ which apply more "aggressive" weighting, such as $\psi(u, v) = |uv|$. Nevertheless, we can apply our theoretical predictions for these choices of $\psi$ after passing the weights through a bounded activation (such as a sigmoid). In Appendix D, we show that our predictions often still show excellent agreement with empirical simulation even when the boundedness assumption is violated.

Our results are stated in terms of the following iteration, for $t \geq 0$:

$$
\begin{aligned}
\tau_{t+1}, \beta_{t+1} &= \arg \max_{\tau \geq 0} \min_{\beta \geq 0} \left\{ \frac{\tau \sigma^2}{\beta} + \tau \beta (1 - \kappa) - \tau^2 + \tau \lambda \, \mathbb{E}_{(V, \Theta) \sim \Pi_t} \left[ \frac{\Theta^2 + \beta^2 \kappa}{\tau V^2 + \beta \lambda} \right] \right\} \\
Q_{t+1} &= \frac{\tau_{t+1} V (\Theta + \beta_{t+1} \sqrt{\kappa} G_t)}{\tau_{t+1} V^2 + \beta_{t+1} \lambda}, \\
\Pi_{t+1} &= \mathrm{Law}(\psi(Q_{t+1}, V), \Theta),
\end{aligned}
\tag{5}
$$

where $G_t \overset{\text{i.i.d.}}{\sim} \mathcal{N}(0, 1)$. In words, given a probability distribution $\Pi_t$ over $\mathbb{R} \times \mathbb{R}$, $\tau_{t+1}$ and $\beta_{t+1}$ are scalars computed as the unique[1] solutions to a *deterministic* optimization problem (this can be solved easily by studying the optimality conditions, as shown in Appendix C). Then, $Q_{t+1}$ is defined as a random variable that is a function of $(V, \Theta) \sim \Pi_t$ and $G_t \sim \mathcal{N}(0, 1)$. Lastly, $\Pi_{t+1}$ is defined as the joint distribution of $\psi(Q_{t+1}, V)$ and $\Theta$.

Given this iteration, we obtain the following result, which is proved in Appendix A:

**Theorem 1.** *Suppose Assumptions 1 and 2 are satisfied. Then, for any $t \geq 0$ and any function $g\colon \mathbb{R}^3 \to \mathbb{R}$ such that $g \in PL(2)$ or $g$ is bounded and continuous, we have*

$$
\frac{1}{d} \sum_{i=1}^{d} g(u_i^{(t+1)}, v_i^{(t)}, \theta_i^*) \overset{P}{\to} \mathbb{E}[g(Q_{t+1}, V, \Theta)],
$$

*where the expectation is over the independent random variables $(V, \Theta) \sim \Pi_t$ and $G_t \sim \mathcal{N}(0, 1)$.*

The limit in this theorem should be interpreted as being the limit as $n, d \to \infty$ with their ratio $\kappa = \frac{d}{n}$ held as a constant. Applying the above theorem for each $t \geq 0$, we can get precise asymptotic predictions for a wide variety of test functions of the iterates. One example of particular interest is the test error, which we measure as the normalized $\ell_1$ distance between $\boldsymbol{u}^{(t+1)} \odot \boldsymbol{v}^{(t)}$ and $\boldsymbol{\theta}^*$, corresponding to $g(u, v, \theta) = |uv - \theta|$ (we provide a proof that this is PL(2) in Proposition 1 in Appendix B). We note that the limiting expectation can be computed via simple Monte Carlo simulation of a *scalar* random variable.

From a technical standpoint, our result is obtained by applying the Convex Gaussian Min-Max Theorem (CGMT) [30, 29] to the weighted least-squares objective function in (4). Previous works have obtained a similar distributional characterization for the solution to least-squares with anisotropic covariates (where the "weights" $\boldsymbol{v}$ are the square root of the eigenvalues of the data covariance) [8]. However, while [8] assume that the eigenvalues are uniformly bounded by constants, this is not a reasonable assumption in our setting, since many common choices of $\psi$ are not bounded and hence

---

[1]The uniqueness of the solution is shown in the proof of Theorem 1.

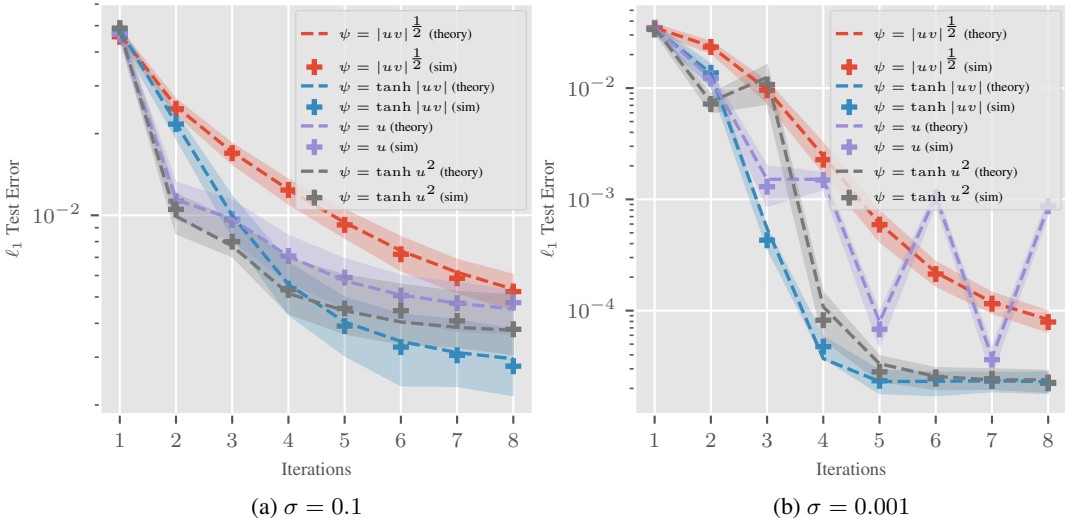

(a) $\sigma = 0.1$          (b) $\sigma = 0.001$

Figure 1: Theoretical predictions and simulations of the test error $\frac{1}{d}\|\boldsymbol{u} \odot \boldsymbol{v} - \boldsymbol{\theta}^*\|_1$ (log scale, pluses denote the median over 100 trials and the shaded region indicates the interquartile range) for two different noise levels, where $n = 250, d = 2000$, and $\boldsymbol{\theta}^*$ has Bernoulli$(0.01)$ entries. Here, $\psi = |uv|^{\frac{1}{2}}$ corresponds to the classical IRLS weighting from [11], $\psi = \tanh |uv|$ is a version of lin-RFM, $\psi = u$ corresponds to AM, and $\psi = \tanh u^2$ is a new reweighting scheme we introduce. We note that the $\psi$ which depend only on $\boldsymbol{u}$ can lead to oscillatory behavior in the test risk.

$\boldsymbol{v}^{(t)}$ is not necessarily bounded uniformly for $t > 1$. A second key difference is that we need to obtain a distributional characterization which can be applied recursively for all $t \geq 0$. The analysis in [8] assumes convergence of the initialization in $\mathcal{W}_4$ distance and proves convergence of the estimator in $\mathcal{W}_3$ distance. However, to apply the result recursively in our setting, if assume that the empirical distribution of $(\boldsymbol{v}^{(0)}, \boldsymbol{\theta}^*)$ converges in $\mathcal{W}_k$ distance, then we need to show that after one iteration, the iterates also converge in $\mathcal{W}_k$ distance (and not in any weaker sense).

To overcome these differences, we use a different technique to show distributional convergence of the iterates. Similar to the approach in [5], we apply the CGMT to a *perturbed* optimization problem, which ultimately allows us to show convergence of test functions of the solutions to the unperturbed problem. While this approach necessitates the additional assumption that $\boldsymbol{\theta}^*$ has bounded entries and we obtain results for a slightly smaller family of test functions $g$ (note, for example, that the squared loss $g(u, v, \theta) = (uv - \theta)^2$ is not PL(2)), we obtain a distributional convergence result that can be applied to a *sequence* of recursively defined least-squares problems which define the trajectory of an algorithm, rather than to a single optimization problem. Moreover, our simulations in Appendix D suggest that the predictions of Theorem 1 still often apply without these additional assumptions, including in the case of the squared loss, indicating that these additional assumptions could potentially be weakened with a more complicated analysis.

### 3.1 Application to sparse linear regression

In this subsection, we apply Theorem 1 to a sparse recovery setting and compare the asymptotic predictions to numerical simulations on high-dimensional Gaussian data. First, we consider a setting where $n = 250, d = 2000$, and $\boldsymbol{\theta}^*$ has Bernoulli$(0.01)$ entries (so the expected sparsity level is $\mathbb{E}[s] = 20$). We run Algorithm 4 with initialization $\boldsymbol{v}^{(0)} = \mathbf{1}_d$ for four different choices of reweighting function and display the test error at each iteration (median over 100 trials) in Figure 1. For each choice of reweighting function $\psi$, we choose the regularization parameter $\lambda$ that minimizes the asymptotic test loss achieved within 8 iterations, and we plot the corresponding trajectory. As shown in the figure, the numerical simulations show excellent alignment with the asymptotic predictions even for this moderate choice of $n$ and $d$.

The asymptotic predictions show that this family of algorithms can find solutions with low test error within only a few iterations. Our results also reveal fine-grained differences in the convergence

behavior of the different algorithms. For instance, more aggressive weightings $\psi = \tanh|uv|$ and $\psi = \tanh u^2$ seem to find better solutions after several iterations. Interestingly, the weighting functions which depend only on $u$ (like alternating minimization) sometimes display a non-monotonic, oscillatory decay of the test loss, particularly in the low-noise regime. However, we do see a steady decrease in test error after every *pair* of iterations (e.g., in AM, after both parameters have been updated). Finally, we note that our framework allows for analysis of new algorithms for training LDNNs. In particular, to our knowledge, weighting functions of the form $\phi(u^2)$ have not been previously considered for this task, but our results indicate that this small modification to AM is competitive with many existing algorithms in this setting.

## 4 Grouped IRLS and the benefits of structured feature learning

In many scenarios, the unknown signal $\boldsymbol{\theta}^*$ is known to possess additional structure that can be leveraged during training. One commonly studied example of this is *structured sparsity*, or group sparsity, where $\boldsymbol{\theta}^*$ has many blocks which are zero. In this section, we generalize the results of Theorem 1 to the case where the reweighting function respects this additional structure in the signal, i.e., $\psi$ acts on blocks of $\boldsymbol{v}$, rather than on individual coordinates.

Concretely, we consider the following modification to our formulation. Let $b \geq 1$ be a constant and write $\mathbb{R}^d$ as a product space over $M = \frac{d}{b}$ factors: $\mathbb{R}^b \times \cdots \times \mathbb{R}^b$. Then, $\boldsymbol{\theta}^*, \boldsymbol{u}^{(t)}, \boldsymbol{v}^{(t)} \in \mathbb{R}^d$ can all be represented as $M$ stacked blocks, each in $\mathbb{R}^b$. Under the same linear measurement model, we now let $\psi : \mathbb{R}^b \times \mathbb{R}^b \to \mathbb{R}^b$ act on each of the *factors* of $(\boldsymbol{u}^{(t)}, \boldsymbol{\theta}^*)$, and consider the same Algorithm 4.

Here, the case $b = 1$ recovers the results of the previous section, but the case $b > 1$ allows us to study the interplay between signal structure and reweighting scheme in a more fine-grained way. For example, suppose $\boldsymbol{\theta}^*$ is known to be group-sparse, meaning that many of the factors $\{\boldsymbol{\theta}_i^*\}_{i=1}^M$ are zero. In this case, it might make sense for $\psi$ to return a vector of the form

$$\psi(\boldsymbol{u}_i^{(t)}, \boldsymbol{v}_i^{(t)}) = \alpha_i \mathbf{1}_b,$$

for some $\alpha_i \in \mathbb{R}$ that is chosen as a function $\boldsymbol{u}_i^{(t)}$ and $\boldsymbol{v}_i^{(t)}$. This corresponds to a reweighting scheme which acts on blocks, rather than individual entries. Another way to motivate this "grouped reweighting" is to leverage the connection to the $\eta$-trick, as in (3). In particular, the group Lasso problem can be written in the following variational form [2]:

$$\min_{\boldsymbol{\theta} \in \mathbb{R}^d} \min_{\boldsymbol{\eta} \in \mathbb{R}_+^M} L(\boldsymbol{\theta}) + \frac{\lambda}{2} \sum_{i=1}^M \left( \frac{\|\boldsymbol{\theta}_i\|_2^2}{\eta_i} + \eta_i \right), \tag{6}$$

where the closed-form solution to the $\boldsymbol{\eta}$ minimization yields the classical group norm regularizer $\sum_{i=1}^M \|\boldsymbol{\theta}_i\|_2$. Here, reparameterizing as $\alpha_i \to \sqrt{\eta_i}$ and $\boldsymbol{u}_i \to \frac{\boldsymbol{\theta}_i}{\sqrt{\eta_i}}$ gives rise naturally to the grouped Hadamard parameterization, with $\boldsymbol{v}_i = \alpha_i \mathbf{1}_b$.

From the perspective of linear diagonal networks, such an approach is equivalent to "tying" together the weights of the hidden layer that correspond to each block. Rather than studying gradient descent/flow for this parameterization (as in [16, 17]), we consider an optimization approach that relies on alternate updates of $\boldsymbol{u}^{(t)}$ and $\boldsymbol{v}^{(t)}$.

We make the same technical assumptions as in Assumptions 1 and 2, with the natural modifications to accommodate $b \geq 1$:

1. $\Pi_0$ is the limit of the empirical distribution of factors of $(\boldsymbol{v}^{(0)}, \boldsymbol{\theta}^*)$ and hence is a distribution over $\mathbb{R}^b \times \mathbb{R}^b$.

2. Each factor $\boldsymbol{\theta}_i^* \in \mathbb{R}^b$ for $i = 1, \ldots, M$ has bounded $\ell_2$-norm almost surely.

3. $\psi$ is bounded and continuous or each of its coordinate projections satisfies $\psi_j^2 \in \mathrm{PL}(2)$ for $j = 1, \ldots, b$.

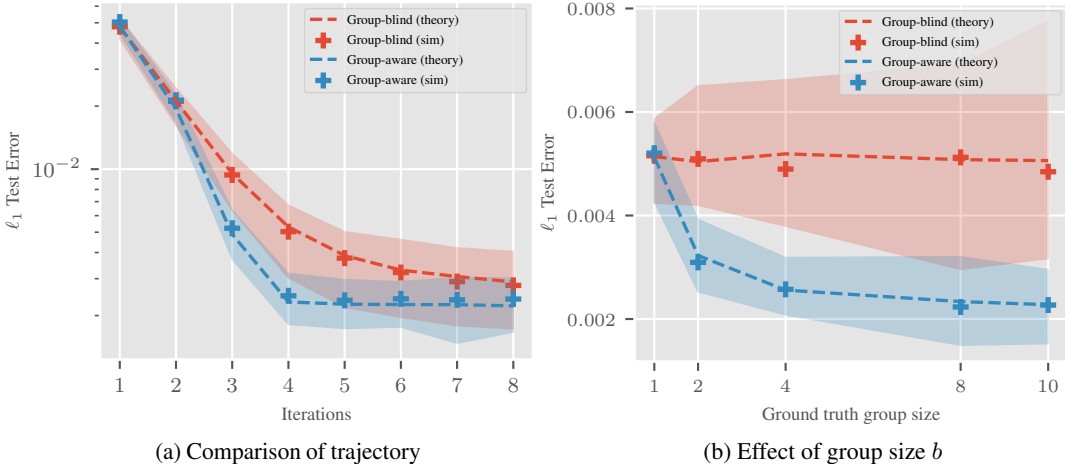

|                                      |                                  |
| :----------------------------------: | :------------------------------: |
| (a) Comparison of trajectory          | (b) Effect of group size $b$     |

Figure 2: Group-blind ($\psi_{gb}$) vs. group-aware ($\psi_{ga}$) reweighting when $\boldsymbol{\theta}^*$ has group-sparse structure. We set $n = 500, d = 4000, \sigma = 0.1$, and $\boldsymbol{\theta}_i^* \overset{\text{i.i.d.}}{\sim} \text{Bernoulli}(0.01)\mathbf{1}_b$. For each curve, $\lambda$ is set to minimize the asymptotic test error achieved. Simulation results are the median/IQR over 100 trials. Left: Comparison of the test error trajectory (log scale) for a fixed block size $b = 8$. Right: $\ell_1$ test error after $T = 4$ iterations, for varying group sizes.

A straightforward extension of Theorem 1 yields the following generalization for the grouped algorithm, where $\boldsymbol{V}, \boldsymbol{\Theta}, \boldsymbol{G}_t, \boldsymbol{Q}_{t+1} \in \mathbb{R}^b$ are now vector-valued random variables: For $t \geq 0$, let

$$
\tau_{t+1}, \beta_{t+1} = \arg\max_{\tau \geq 0} \min_{\beta \geq 0} \left\{ \frac{\tau\sigma^2}{\beta} + \tau\beta(1-\kappa) - \tau^2 + \tau\lambda \, \mathbb{E}_{(\boldsymbol{V},\boldsymbol{\Theta})\sim\Pi_t} \left[ \frac{1}{b} \sum_{j=1}^{b} \frac{\Theta_j^2 + \beta^2\kappa}{\tau V_j^2 + \beta\lambda} \right] \right\}
$$

$$
\boldsymbol{Q}_{t+1} = \frac{\tau_{t+1}\boldsymbol{V} \odot (\boldsymbol{\Theta} + \beta_{t+1}\boldsymbol{G}_t\sqrt{\kappa})}{\tau_{t+1}\boldsymbol{V}^{\odot 2} + \beta_{t+1}\lambda\mathbf{1}_b}, \quad \text{(entry-wise division)}
$$

$$
\Pi_{t+1} = \text{Law}(\psi(\boldsymbol{Q}_{t+1}, \boldsymbol{V}), \boldsymbol{\Theta}).
$$

(7)

Here, $\boldsymbol{G}_t \overset{\text{i.i.d.}}{\sim} \mathcal{N}(\mathbf{0}, \boldsymbol{I}_b)$. Then, we have the following result, which is proved in Appendix A.

**Theorem 2.** *[Generalization of Theorem 1 for $b \geq 1$] Under the assumptions above, for any $t \geq 0$ and any function $g\colon (\mathbb{R}^b)^3 \to \mathbb{R}$ such that $g \in PL(2)$ or $g$ is bounded and continuous, we have*

$$
\frac{1}{M} \sum_{i=1}^{M} g(\boldsymbol{u}_i^{(t+1)}, \boldsymbol{v}_i^{(t)}, \boldsymbol{\theta}_i^*) \overset{P}{\to} \mathbb{E}[g(\boldsymbol{Q}_{t+1}, \boldsymbol{V}, \boldsymbol{\Theta})].
$$

Given a reweighting function $\psi$, Theorem 2 characterizes the distribution of the factors (blocks) of the iterates. Hence, by choosing $g(\boldsymbol{u}, \boldsymbol{v}, \boldsymbol{\theta}) = |\boldsymbol{u} \odot \boldsymbol{v} - \boldsymbol{\theta}|$, we can predict the exact limiting test error for this family of algorithms.

Computing these theoretical predictions reveals that choosing $\psi$ in a group-aware way can lead to significant performance improvements compared to coordinate-wise reweighting. In Figure 2, we fix $\sigma = 0.1, n = 500, d = 4000$, and set the overall expected sparsity level of $\boldsymbol{\theta}^*$ as in Figure 1. We compare the performance of Algorithm 4 for a "group-blind" ($\psi_{gb}$) and "group-aware" ($\psi_{ga}$) choice of reweighting function:

- $\psi_{gb}(\boldsymbol{u}, \boldsymbol{v}) = \tanh|\boldsymbol{u} \odot \boldsymbol{v}|$ — note this is identical to one of the reweightings considered in Section 3.1.

- $\psi_{ga}(\boldsymbol{u}, \boldsymbol{v}) = \left(\frac{1}{b} \sum_{j=1}^{b} \tanh|u_j v_j|\right)\mathbf{1}_b$

The theoretical predictions align with simulations and show a notable improvement in performance when using the group-aware scheme with $b > 1$. Moreover, as the group size $b$ increases, the performance of $\psi_{gb}$ remains approximately the same, indicating that it is not able to take adapt to the

group-structure. By contrast, using $\psi_{ga}$ leads to a consistent improvement in test error as $b$ gets larger. Hence, the test error when using the group-aware scheme scales with the size/number of groups, rather than the overall sparsity level.

## 5   Conclusion

In this paper, we derived a precise asymptotic characterization of the iterates of a family of algorithms for learning high-dimensional linear models with linear diagonal networks. We used these predictions to obtain fine-grained predictions of the test error at each iteration for various existing algorithms for this task, and we showed that our framework can also be used as a test bed for new variations on these algorithms that take a similar form. Lastly, we demonstrated the advantage of embedding more structure into the model by tying together groups of weights when the ground-truth has structured sparsity. Several interesting open questions about these types of algorithms remain. While our simulations align very well with the predicted asymptotic trajectory, it would be interesting to obtain finite-sample guarantees that hold even for batch sizes that are much smaller than $d$ (as in the "mini-batch" case studied by [19]). Moreover, seeing as our analysis depends crucially on the independence the covariates at every iteration, developing precise predictions of the trajectory in the non-batched setting remains an interesting direction for future work.

## Acknowledgments and Disclosure of Funding

We thank the anonymous reviewers for their helpful feedback. This work was supported by an NSF Graduate Research Fellowship (DGE-2039655), the NSF AI Institute AI4OPT, NSF grants IIS-2212182, CCF-223915 and 2112533, and gifts from Amazon and Adobe.

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

# A Proof of main results

In this section, we provide the proofs of our main results.

## A.1 Notation and background

For convenience, we first restate the main notation that is used in our proofs.

**Notation** The ones vector of dimension $d$ is denoted $\mathbf{1}_d$. We write $a \lesssim b$ when $a \leq Cb$ for some sufficiently large constant $C > 0$ which does not depend on $d$. We denote the element-wise multiplication (Hadamard product) of two vectors $\boldsymbol{x}$ and $\boldsymbol{y}$ as $\boldsymbol{x} \odot \boldsymbol{y}$. Element-wise division of two vectors is denoted as $\frac{\boldsymbol{x}}{\boldsymbol{y}}$. We use the shorthand $(\cdot)_+ = \max(\cdot, 0)$. A function $f \colon \mathbb{R}^p \to \mathbb{R}$ is called *pseudo-Lipschitz* of order 2 if, for all $\boldsymbol{x}, \boldsymbol{y} \in \mathbb{R}^p$,

$$|f(\boldsymbol{x}) - f(\boldsymbol{y})| \leq C(1 + \|\boldsymbol{x}\|_2 + \|\boldsymbol{y}\|_2)\|\boldsymbol{x} - \boldsymbol{y}\|_2$$

for some constant $C > 0$. The set of such functions is denoted PL(2).

Convergence in probability of a sequence of random variables $X_d$ to a random variable $X$ is denoted $X_d \xrightarrow{P} X$. Convergence in Wasserstein-2 distance of a sequence of probability distributions $\nu_d$ to a limiting distribution $\nu$ is denoted as $\nu_d \xrightarrow{\mathcal{W}_2} \nu$, and this fact is equivalent to the statement $\mathbb{E}_{X \sim \nu_d} g(X) \to \mathbb{E}_{X \sim \nu} g(X)$ for all $g \in$ PL(2) [3]. If the $\nu_d$ are *random* probability measures, we say that $\nu_d \xrightarrow{\mathcal{W}_2} \nu$ if the same convergence holds in probability, i.e., $\mathbb{E}_{X \sim \nu_d} g(X) \xrightarrow{P} \mathbb{E}_{X \sim \nu} g(X)$ for all $g \in$ PL(2). The empirical distribution of a vector $\boldsymbol{z} \in \mathbb{R}^d$ is defined as $\frac{1}{d} \sum_{i=1}^{d} \delta(z_i)$, where $\delta(z_i)$ is the Dirac delta distribution centered at $z_i$. For any random variable (or group of random variables) $X$, we use the notation $\mathrm{Law}(X)$ to denote the probability distribution of $X$.

We also define two key quantities which appear in the analysis.

**Definition 1.** *The Moreau envelope function of a lower semi-continuous, proper convex function $\ell \colon \mathbb{R}^p \to \mathbb{R}$ with step size $\tau$ is defined as*

$$\mathcal{M}_\ell(\boldsymbol{x}; \tau) = \min_{\boldsymbol{y} \in \mathbb{R}^p} \ell(\boldsymbol{y}) + \frac{1}{2\tau} \|\boldsymbol{y} - \boldsymbol{x}\|_2^2.$$

*The proximal (prox) operator of $\ell$ with step size $\tau$, denoted $\mathrm{prox}_\ell(\boldsymbol{x}, \tau)$ is defined as the $\arg\min$ of the above optimization problem.*

Lastly, we restate the version of the Convex Gaussian Min-Max Theorem (CGMT) that we will use in our proofs.

**Theorem 3** (Convex Gaussian Min-Max Theorem [29]). *Let $\boldsymbol{G} \in \mathbb{R}^{m \times n}, \boldsymbol{g} \in \mathbb{R}^m, \boldsymbol{h} \in \mathbb{R}^n$ have i.i.d. $\mathcal{N}(0, 1)$ entries. Let $\mathcal{S}_w \subset \mathbb{R}^n$ and $\mathcal{S}_u \subset \mathbb{R}^m$ be compact, convex sets, and $f \colon \mathbb{R}^n \times \mathbb{R}^m \to \mathbb{R}$ be convex-concave on $\mathcal{S}_w \times \mathcal{S}_u$. Define the following two min-max problems:*

$$\Phi(\boldsymbol{G}) := \min_{\boldsymbol{w} \in \mathcal{S}_w} \max_{\boldsymbol{u} \in \mathcal{S}_u} \boldsymbol{u}^\top \boldsymbol{G} \boldsymbol{w} + f(\boldsymbol{w}, \boldsymbol{u})$$

$$\phi(\boldsymbol{g}, \boldsymbol{h}) := \min_{\boldsymbol{w} \in \mathcal{S}_w} \max_{\boldsymbol{u} \in \mathcal{S}_u} \|\boldsymbol{w}\|_2 \boldsymbol{u}^\top \boldsymbol{g} + \|\boldsymbol{u}\|_2 \boldsymbol{w}^\top \boldsymbol{h} + f(\boldsymbol{w}, \boldsymbol{u})$$

*Then, for all $c \in \mathbb{R}$ and $t > 0$,*

$$\mathbb{P}\{|\Phi(\boldsymbol{G}) - c| > t\} \leq 2\,\mathbb{P}\{|\phi(\boldsymbol{g}, \boldsymbol{h}) - c| > t\}$$

## A.2 Proof of Theorems 1 and 2

*Proof of Theorem 1.* Assume that $\frac{1}{d} \sum_{i=1}^{d} \delta(v_i^{(t)}, \theta_i^*) \xrightarrow{\mathcal{W}_2} \Pi_t$ (note that this holds by assumption at $t = 0$; we will show later that it holds at time $t + 1$).

First observe that convergence of the joint empirical distribution of $(\boldsymbol{u}^{(t+1)}, \boldsymbol{v}^{(t)}, \boldsymbol{\theta}^*)$ to the joint distribution of $(Q_{t+1}, V, \Theta)$ in Wasserstein-2 distance implies that

$$\frac{1}{d} \sum_{i=1}^{d} g(u_i^{(t+1)}, v_i^{(t)}, \theta_i^*) \xrightarrow{P} \mathbb{E}[g(Q_{t+1}, V, \Theta)],$$

for any $g \in \text{PL}(2)$ or which is bounded and continuous. This is because $\mathcal{W}_2$ convergence implies convergence in expectation of any pseudo-Lipschitz function of order 2 [3, Lemma 5] and of any bounded continuous function (since $\mathcal{W}_2$ convergence is stronger than weak convergence [33, Theorem 6.9]). Hence, it suffices to show that

$$\frac{1}{d} \sum_{i=1}^{d} \delta(u_i^{(t+1)}, v_i^{(t)}, \theta_i^*) \overset{\mathcal{W}_2}{\to} \text{Law}(Q_{t+1}, V, \Theta), \tag{8}$$

where $(V, \Theta) \sim \Pi_t$ and $Q_{t+1}$ is defined as in Eq. 5.

Recall that the objective function for the update on $\boldsymbol{u}$ is given by

$$\boldsymbol{u}^{(t+1)} = \arg\min_{\boldsymbol{u} \in \mathbb{R}^d} \frac{1}{n} \left\| \boldsymbol{y}^{(t)} - \frac{1}{\sqrt{d}} \boldsymbol{X}^{(t)}(\boldsymbol{u} \odot \boldsymbol{v}^{(t)}) \right\|_2^2 + \frac{\lambda}{d} \|\boldsymbol{u}\|_2^2.$$

Rather than study this update directly, we first analyze a slightly more general problem (following the approach in [5]). Let $h \colon \mathbb{R}^3 \to \mathbb{R}$ be a continuous test function with $\|\nabla^2 h\|_2 \leq C$ and that satisfies one of the following:

1. $h$ is uniformly bounded.

2. $h(u, v, \theta) = u^2$.

Then we consider the following problem (the dependence of $\boldsymbol{X}, \boldsymbol{y}, \boldsymbol{\epsilon}$, and $\boldsymbol{v}$ on $t$ is dropped to simplify the notation):

$$P_1(\mu) = \min_{\boldsymbol{u} \in \mathbb{R}^d} \frac{1}{n} \left\| \boldsymbol{y} - \frac{1}{\sqrt{d}} \boldsymbol{X}(\boldsymbol{u} \odot \boldsymbol{v}) \right\|_2^2 + \frac{\lambda}{d} \|\boldsymbol{u}\|_2^2 + \frac{\mu}{d} \sum_{i=1}^{d} h(u_i, v_i, \theta_i^*), \tag{9}$$

where $\mu \in [-\mu^*, \mu^*]$ and $\mu^* = \frac{\lambda}{C}$ is chosen sufficiently small so that the objective function (scaled by $d$) is $\lambda$-strongly convex for all $\mu$ in this range. The case $\mu = 0$ recovers the original problem of interest.

**Step 1: Convergence of the loss** Rewriting this in terms of the error vector $\boldsymbol{\Delta} := \frac{1}{\sqrt{d}}(\boldsymbol{u} \odot \boldsymbol{v} - \boldsymbol{\theta}^*)$, we have

$$P_1(\mu) = \min_{\boldsymbol{\Delta} \in \mathbb{R}^d} \frac{1}{n} \|\boldsymbol{\epsilon} - \boldsymbol{X}\boldsymbol{\Delta}\|_2^2 + \frac{\lambda}{d} \left\| \frac{\sqrt{d}\boldsymbol{\Delta} + \boldsymbol{\theta}^*}{\boldsymbol{v}} \right\|_2^2 + \frac{\mu}{d} \sum_{i=1}^{d} h\left( \frac{\sqrt{d}\Delta_i + \theta_i^*}{v_i}, v_i, \theta_i^* \right).$$

In writing this, we use the fact that $v_i \neq 0$ for all $i$ with probability 1 (and the notation in the second-to-last term indicates entry-wise division). Now, using the identity $\|\cdot\|_2^2 = \max_{\boldsymbol{q}} 2\boldsymbol{q}^\top(\cdot) - \|\boldsymbol{q}\|_2^2$, we can write this as

$$P_1(\mu) = \min_{\boldsymbol{\Delta} \in \mathbb{R}^d} \max_{\boldsymbol{q} \in \mathbb{R}^n} \frac{2}{\sqrt{n}} \boldsymbol{q}^\top \boldsymbol{\epsilon} - \frac{2}{\sqrt{n}} \boldsymbol{q}^\top \boldsymbol{X}\boldsymbol{\Delta} - \|\boldsymbol{q}\|_2^2 + \frac{\lambda}{d} \left\| \frac{\sqrt{d}\boldsymbol{\Delta} + \boldsymbol{\theta}^*}{\boldsymbol{v}} \right\|_2^2 + \frac{\mu}{d} \sum_{i=1}^{d} h\left( \frac{\sqrt{d}\Delta_i + \theta_i^*}{v_i}, v_i, \theta_i^* \right). \tag{10}$$

Next, in Lemma 1, we show that there exist Euclidean balls $B_\Delta$ and $B_q$, each of radius $C_1\|\boldsymbol{v}\|_\infty$ such that, with probability approaching 1, we can constrain the feasible set to lie with these balls without changing the value of $P_1(\mu)$, so we can study

$$\tilde{P}_1(\mu) = \min_{\boldsymbol{\Delta} \in B_\Delta} \max_{\boldsymbol{q} \in B_q} \frac{2}{\sqrt{n}} \boldsymbol{q}^\top \boldsymbol{\epsilon} - \frac{2}{\sqrt{n}} \boldsymbol{q}^\top \boldsymbol{X}\boldsymbol{\Delta} - \|\boldsymbol{q}\|_2^2 + \frac{\lambda}{d} \left\| \frac{\sqrt{d}\boldsymbol{\Delta} + \boldsymbol{\theta}^*}{\boldsymbol{v}} \right\|_2^2 + \frac{\mu}{d} \sum_{i=1}^{d} h\left( \frac{\sqrt{d}\Delta_i + \theta_i^*}{v_i}, v_i, \theta_i^* \right), \tag{11}$$

where $P_1(\mu) = \tilde{P}_1(\mu)$ with probability tending to 1. We can therefore condition on this event for the remainder of the analysis without changing our asymptotic conclusions.

Now, noting that this is in the correct form to apply Theorem 3, we define the auxiliary optimization problem

$$P_2(\mu) = \min_{\boldsymbol{\Delta} \in B_\Delta} \max_{\boldsymbol{q} \in B_q} \frac{2}{\sqrt{n}} \boldsymbol{q}^\top \boldsymbol{\epsilon} - \frac{2}{\sqrt{n}} \|\boldsymbol{q}\|_2 \boldsymbol{g}^\top \boldsymbol{\Delta} - \frac{2}{\sqrt{n}} \|\boldsymbol{\Delta}\|_2 \boldsymbol{h}^\top \boldsymbol{q} - \|\boldsymbol{q}\|_2^2 + \frac{\lambda}{d} \left\| \frac{\sqrt{d}\boldsymbol{\Delta} + \boldsymbol{\theta}^*}{\boldsymbol{v}} \right\|_2^2 \tag{12}$$

$$+ \frac{\mu}{d} \sum_{i=1}^{d} h\left( \frac{\sqrt{d}\Delta_i + \theta_i^*}{v_i}, v_i, \theta_i^* \right), \tag{13}$$

where $\boldsymbol{g} \in \mathbb{R}^d$ and $\boldsymbol{h} \in \mathbb{R}^n$ have i.i.d. standard normal entries. By Theorem 3, for all $\delta > 0$ and fixed $\bar{P}(\mu) \in \mathbb{R}$,

$$\mathbb{P}\{|P_1(\mu) - \bar{P}(\mu)| > \delta\} \leq 2\,\mathbb{P}\{|P_2(\mu) - \bar{P}(\mu)| > \delta\}.$$

In particular, if we can find some $\bar{P}(\mu)$ such that $P_2(\mu) \xrightarrow{P} \bar{P}(\mu)$, then we can conclude also that $P_1(\mu) \xrightarrow{P} \bar{P}(\mu)$.

To accomplish this, we next perform a series of simplifications to $P_2(\mu)$ which will later help us characterize its asymptotic behavior. First, we can decouple the optimization over $\boldsymbol{q}$ into its norm and direction, and the latter can be solved explicitly. Letting $\tau = \|\boldsymbol{q}\|_2$, this yields

$$P_2(\mu) = \min_{\boldsymbol{\Delta} \in B_\Delta} \max_{0 \leq \tau \leq R} \frac{2\tau}{\sqrt{n}} \|\boldsymbol{\epsilon} - \|\boldsymbol{\Delta}\|_2 \boldsymbol{h}\|_2 - \frac{2\tau}{\sqrt{n}} \boldsymbol{g}^\top \boldsymbol{\Delta} - \tau^2 + \frac{\lambda}{d} \left\| \frac{\sqrt{d}\boldsymbol{\Delta} + \boldsymbol{\theta}^*}{\boldsymbol{v}} \right\|_2^2 \tag{14}$$

$$+ \frac{\mu}{d} \sum_{i=1}^{d} h\left( \frac{\sqrt{d}\Delta_i + \theta_i^*}{v_i}, v_i, \theta_i^* \right),$$

where $R \coloneqq C_1 \|\boldsymbol{v}\|_\infty$. Next, note that $\boldsymbol{\epsilon}$ and $\boldsymbol{h}$ are independent Gaussian vectors and hence $\boldsymbol{\epsilon} - \|\boldsymbol{\Delta}\|_2 \boldsymbol{h} \stackrel{d}{=} \sqrt{\sigma^2 + \|\boldsymbol{\Delta}\|_2^2}\,\boldsymbol{h}$. So, we have

$$P_2(\mu) \stackrel{d}{=} \min_{\boldsymbol{\Delta} \in B_\Delta} \max_{0 \leq \tau \leq R} \frac{2\tau \|\boldsymbol{h}\|_2}{\sqrt{n}} \sqrt{\sigma^2 + \|\boldsymbol{\Delta}\|_2^2} - \frac{2\tau}{\sqrt{n}} \boldsymbol{g}^\top \boldsymbol{\Delta} - \tau^2 + \frac{\lambda}{d} \left\| \frac{\sqrt{d}\boldsymbol{\Delta} + \boldsymbol{\theta}^*}{\boldsymbol{v}} \right\|_2^2$$

$$+ \frac{\mu}{d} \sum_{i=1}^{d} h\left( \frac{\sqrt{d}\Delta_i + \theta_i^*}{v_i}, v_i, \theta_i^* \right). \tag{15}$$

Before proceeding further, we rewrite this in terms of a minimization over the variable $\boldsymbol{u} = \frac{\sqrt{d}\boldsymbol{\Delta} + \boldsymbol{\theta}^*}{\boldsymbol{v}}$.

$$P_2(\mu) \stackrel{d}{=} \min_{\boldsymbol{u} \in B_u} \max_{0 \leq \tau \leq R} \frac{2\tau \|\boldsymbol{h}\|_2}{\sqrt{n}} \sqrt{\sigma^2 + \frac{1}{d}\|\boldsymbol{u} \odot \boldsymbol{v} - \boldsymbol{\theta}^*\|_2^2} - \frac{2\tau}{\sqrt{nd}} \boldsymbol{g}^\top (\boldsymbol{u} \odot \boldsymbol{v} - \boldsymbol{\theta}^*)$$

$$- \tau^2 + \frac{\lambda}{d} \|\boldsymbol{u}\|_2^2 + \frac{\mu}{d} \sum_{i=1}^{d} h(u_i, v_i, \theta_i^*) \tag{16}$$

Here, $B_u \coloneqq \left\{ \boldsymbol{u} \in \mathbb{R}^d \colon \left\| \frac{1}{\sqrt{d}} (\boldsymbol{u} \odot \boldsymbol{v} - \boldsymbol{\theta}^*) \right\|_2 \leq R \right\}$. After this step, observe that the objective function is strongly concave in $\tau$ and strongly convex in $\boldsymbol{u}$ (the sum of the last two terms is strongly convex in $\boldsymbol{u}$ based on the assumption that $h$ has bounded Hessian and $\mu$ is sufficiently small). Since the objective function is convex-concave over convex and compact sets, we can invoke Sion's minimax theorem to switch the min and max. Furthermore, we use the fact that $\sqrt{x} = \min_{\beta > 0} \frac{x}{2\beta} + \frac{\beta}{2}$ to write this as

$$P_2(\mu) \stackrel{d}{=} \max_{0 \leq \tau \leq R} \min_{\substack{\boldsymbol{u} \in B_u, \\ \sigma \leq \beta \leq \sigma + R}} \frac{\tau \|\boldsymbol{h}\|_2}{\sqrt{n}} \left( \frac{\sigma^2}{\beta} + \frac{\|\boldsymbol{u} \odot \boldsymbol{v} - \boldsymbol{\theta}^*\|_2^2}{\beta d} + \beta \right) - \frac{2\tau}{\sqrt{nd}} \boldsymbol{g}^\top (\boldsymbol{u} \odot \boldsymbol{v} - \boldsymbol{\theta}^*)$$

$$- \tau^2 + \frac{\lambda}{d} \|\boldsymbol{u}\|_2^2 + \frac{\mu}{d} \sum_{i=1}^{d} h(u_i, v_i, \theta_i^*). \tag{17}$$

Here, note that we can add the constraint on $\beta$ without changing the solution since the optimal value of $\beta$ will be obtained at $\sqrt{\sigma^2 + \frac{1}{d}\|\boldsymbol{u} \odot \boldsymbol{v} - \boldsymbol{\theta}^*\|_2^2} \in [\sigma, \sigma + R]$ for all feasible $\boldsymbol{u}$.

Next, we can explicitly solve the inner minimization over $\boldsymbol{u}$. To do this, we first show in Lemma 2 that the optimal solution to the unconstrained minimization is strictly feasible for large enough $C_1$, and hence, the unconstrained and constrained minimizations over $\boldsymbol{u}$ are equivalent. Next, observe that the unconstrained problem is separable over the indices, so we need only to solve the scalar problem

$$\min_{u_i \in \mathbb{R}} \frac{\tau\|\boldsymbol{h}\|_2(u_iv_i - \theta_i^*)^2}{\beta d\sqrt{n}} - \frac{2\tau}{\sqrt{nd}}g_i(u_iv_i - \theta_i^*) + \frac{\lambda}{d}u_i^2 + \frac{\mu}{d}h(u_i, v_i, \theta_i^*) \tag{18}$$

Completing the squares, we obtain that the above problem can be written in terms of the Moreau envelope (Definition 1) of the function $\ell(u) = \lambda u^2 + \mu h$:

$$\frac{1}{d}\left[\frac{\tau\xi\theta_i^{*2}}{\beta} - \frac{\tau}{\beta\xi}\left(\beta g_i\sqrt{\kappa} - \xi\theta_i^*\right)^2 + \mathcal{M}_{\lambda(\cdot)^2 + \mu h(\cdot, v_i, \theta_i^*)}\left(\frac{\xi\theta_i^* - \beta g_i\sqrt{\kappa}}{\xi v_i}; \frac{\beta}{2\xi\tau v_i^{(t)2}}\right)\right],$$

where we have introduced the shorthand notation $\xi = \frac{\|\boldsymbol{h}\|_2}{\sqrt{n}}$. Substituting this into the expression for $P_2$ above, we obtain

$$P_2(\mu) \stackrel{d}{=} \max_{0 \le \tau \le R} \min_{\sigma \le \beta \le \sigma + R} \frac{\tau\sigma^2\xi}{\beta} \quad + \tau\beta\xi - \tau^2 + \frac{1}{d}\sum_{i=1}^d\left[\frac{\tau\xi\theta_i^{*2}}{\beta} - \frac{\tau}{\beta\xi}\left(\beta g_i\sqrt{\kappa} - \xi\theta_i^*\right)^2\right]$$

$$+ \frac{1}{d}\sum_{i=1}^d\left[\mathcal{M}_{\lambda(\cdot)^2 + \mu h(\cdot, v_i, \theta_i^*)}\left(\frac{\xi\theta_i^* - \beta g_i\sqrt{\kappa}}{\xi v_i}; \frac{\beta}{2\xi\tau v_i^{(t)2}}\right)\right]$$

$$:= \max_{0 \le \tau \le R} \min_{\sigma \le \beta \le \sigma + R} f_d(\tau, \beta) \tag{19}$$

Now that the optimization has been fully "scalarized", we proceed by considering its asymptotic behavior. First, note that the partial minimization over $\boldsymbol{u}$ preserves the concavity/convexity in $(\tau, \beta)$. In Lemma 3, we prove that for any fixed $\tau$ and $\beta$, the objective function $f_d(\tau, \beta)$ converges in probability to

$$f(\tau, \beta) = \frac{\tau\sigma^2}{\beta} + \tau\beta(1 - \kappa) - \tau^2 + \mathbb{E}\left[\mathcal{M}_{\lambda(\cdot)^2 + \mu h(\cdot, V, \Theta)}\left(\frac{\Theta - \beta G\sqrt{\kappa}}{V}; \frac{\beta}{2\tau V^2}\right)\right], \tag{20}$$

where the expectation is over $(V, \Theta) \sim \Pi_t$ and an independent $G \sim \mathcal{N}(0, 1)$. We note here that Lemma 3 is the only place in our proof which requires the boundedness of the entries of $\boldsymbol{\theta}^*$.

Since $f_d(\tau, \beta)$ is strongly concave in $\tau$ with parameter 1 for all feasible $\beta$, we can conclude that $f(\tau, \beta)$ is also strongly concave in $\tau$ with parameter 1. Directly taking a derivative with respect to $\beta$, we also find that $f$ has a single non-negative critical point, at the point $\hat{\beta} = \sqrt{\sigma^2 + \mathbb{E}[(\hat{u}V - \Theta)^2]}$, where

$$\hat{u} = \hat{u}(V, \Theta) = \text{prox}_{\lambda(\cdot)^2 + \mu h(\cdot, V, \Theta)}\left(\frac{\Theta - \beta G\sqrt{\kappa}}{V}; \frac{\beta}{2\tau V^2}\right).$$

So, we can conclude that $f$ has unique saddle point $(\hat{\tau}, \hat{\beta})$. Note $f$ is a deterministic function that does not depend on $d$ and hence $(\hat{\tau}, \hat{\beta})$ are also deterministic and independent of $d$.

Now let $C_2 := \max\{\hat{\tau}, \hat{\beta}\} + 1$. By the "convexity lemma" (as stated in [24]), pointwise convergence (in probability) of a convex function is uniform over compact sets. So, this result implies that the convergence is uniform over $(\tau, \beta) \in [0, C_2] \times [0, C_2],$[2] so

$$\max_{0 \le \tau \le C_2} \min_{0 \le \beta \le C_2} f_d(\tau, \beta) \stackrel{P}{\to} \max_{0 \le \tau \le C_2} \min_{0 \le \beta \le C_2} f(\tau, \beta)$$

Let $(\hat{\tau}_d, \hat{\beta}_d)$ denote the optimnal solution for the problem on the left. We can also conclude that $(\hat{\tau}_d, \hat{\beta}_d) \stackrel{P}{\to} (\hat{\tau}, \hat{\beta})$ by [22, Theorem 2.1], which states that uniform convergence in probability of

---

[2]Note we could not have directly applied this to the feasible sets of $P_2(\mu)$, since $R$ may have a dependence on $d$.

a convex function over a compact set implies convergence of the optimal minimizer. So, with probability approaching 1, $(\hat{\tau}_d, \hat{\beta}_d)$ are strictly smaller than $C_2$, and the same solution is also optimal for $P_2(\mu)$. We can therefore conclude

$$P_2(\mu) \xrightarrow{P} \max_{0 \le \tau \le C_2} \min_{0 \le \beta \le C_2} f(\tau, \beta) = \max_{\tau \ge 0} \min_{\beta \ge 0} f(\tau, \beta) =: \bar{P}(\mu)$$

Therefore, by Theorem 3, for any fixed $\mu \in [-\mu^*, \mu^*]$, we have the convergence

$$P_1(\mu) \xrightarrow{P} \bar{P}(\mu).$$

In the special case $\mu = 0$, we can further simplify the Moreau envelope term to obtain

$$\bar{P}(0) := \max_{\tau \ge 0} \min_{\beta \ge 0} \frac{\tau \sigma^2}{\beta} + \tau\beta(1 - \kappa) - \tau^2 + \tau\lambda \mathbb{E}\left[\frac{\Theta^2 + \beta^2\kappa}{\tau V^2 + \beta\lambda}\right]. \tag{21}$$

**Step 2: Convergence of the optimal solution** We next need to extend this result to the desired Wasserstein-2 convergence result (8). Recall here that $\boldsymbol{u}^{(t+1)}$ is the solution of $P_1(0)$.

First, let $h \colon \mathbb{R}^3 \to \mathbb{R}$ be any bounded, Lipschitz function, and let $h^{(k)}$ be a sequence of bounded, twice-differentiable functions that converge uniformly to $h$ as $k \to \infty$ (e.g., the convolution of $h$ with a sequence of mollifiers). Let $P_1^{(k)}(\mu), \bar{P}^{(k)}(\mu)$ be the optimal cost of $P_1$ and $\bar{P}$ when using test function $h^{(k)}$ and for $\mu \in [-\mu^*, \mu^*]$. Note the convergence $P_1^{(k)}(\mu) \xrightarrow{P} \bar{P}^{(k)}(\mu)$ for any $\mu$ in a sufficiently small neighborhood around zero holds by Step 1.

By the uniform convergence of the $h^{(k)}$ to $h$,

$$\lim_{k\to\infty} P_1^{(k)}(\mu) = P_1(\mu)$$

$$\lim_{k\to\infty} \bar{P}^{(k)}(\mu) = \bar{P}(\mu).$$

Now, fix $\delta > 0$ and choose $k$ large enough that $|P_1^{(k)}(\mu) - P_1(\mu)| < \frac{\delta}{3}$ and $|\bar{P}^{(k)}(\mu) - \bar{P}(\mu)| < \frac{\delta}{3}$. Then,

$$\mathbb{P}\left\{|P_1(\mu) - \bar{P}(\mu)| > \delta\right\} \le \mathbb{P}\left\{|P_1^{(k)}(\mu) - \bar{P}^{(k)}(\mu)| > \delta/3\right\} \to 0,$$

since $P_1^{(k)} \xrightarrow{P} \bar{P}_2^{(k)}$ for all $k$. Hence, we can also apply the result of Step 1 to any bounded Lipschitz function $h$.

Since the convergence result of Step 1 holds for any $\mu$ in a neighborhood around zero, we can conclude that

$$\frac{1}{d}\sum_{i=1}^d h(u_i^{(t+1)}, v_i, \theta_i^*) \xrightarrow{P} \frac{d\bar{P}(\mu)}{d\mu}\bigg|_{\mu=0},$$

where the derivative is well-defined since $\bar{P}$ has a unique solution in a neighborhood around zero. The proof of this fact is identical to that of Lemma 7 of [5], so we omit it here. Moreover, using the Dominated Convergence Theorem to differentiate inside the expectation, we can compute this exactly:

$$\frac{d\bar{P}(\mu)}{d\mu}\bigg|_{\mu=0} = \mathbb{E}\, h\left(\mathrm{prox}_{\lambda(\cdot)^2}\left(\frac{\Theta - \hat{\beta} G\sqrt{\kappa}}{V}; \frac{\hat{\beta}}{2\hat{\tau}V^2}\right), V, \Theta\right) = \mathbb{E}\, h\left(\frac{\hat{\tau}V(\Theta - \hat{\beta} G\sqrt{\kappa})}{\hat{\tau}V^2 + \hat{\beta}\lambda}, V, \Theta\right),$$

where $(\hat{\beta}, \hat{\tau})$ are found in the optimal solution to $\bar{P}(0)$.

Hence, for all bounded, Lipschitz $h$, we have

$$\frac{1}{d}\sum_{i=1}^d h(u_i^{(t+1)}, v_i, \theta_i^*) \xrightarrow{P} \mathbb{E}\, h\left(\frac{\hat{\tau}V(\Theta - \hat{\beta} G\sqrt{\kappa})}{\hat{\tau}V^2 + \hat{\beta}\lambda}, V, \Theta\right),$$

so the empirical distribution of the triple $(u_i^{(t+1)}, v_i, \theta_i^*)$ converges weakly to the distribution of the random variable $\left(\frac{\hat{\tau}V(\Theta - \hat{\beta} G\sqrt{\kappa})}{\hat{\tau}V^2 + \hat{\beta}\lambda}, V, \Theta\right)$, where $G \sim \mathcal{N}(0,1)$ and $(V, \Theta) \sim \Pi_t$. By choosing $h(u, v, \theta) = u^2$, we also know that second moments of the empirical distribution converge in probability. Hence, the convergence can be strengthened from weak convergence to convergence in $\mathcal{W}_2$ distance (see, e.g. [33, Theorem 6.9]).

**Step 3: Verifying the inductive hypothesis**   Lastly, we need to show that

$$\frac{1}{d}\sum_{i=1}^{d}\delta(v_i^{(t+1)},\theta_i^*) \overset{\mathcal{W}_2}{\to} \mathrm{Law}(\psi(Q_{t+1},V),\Theta) := \Pi_{t+1}.$$

where $(V,\Theta)\sim\Pi_t$. Here, weak convergence follows from the result of Step 2 since $\psi$ is a continuous map. To show convergence of second moments, we need to show

$$\frac{1}{d}\sum_{i=1}^{d}v_i^{(t+1),2} = \frac{1}{d}\sum_{i=1}^{d}\psi(u_i^{(t+1)},v_i^{(t)})^2 \overset{P}{\to} \mathbb{E}[\psi(Q_{t+1},V)^2].$$

For $\psi$ that satisfies Assumption 2, this convergence is immediate from the result of Step 2 (since $\mathcal{W}_2$ convergence implies convergence in expectation of bounded continuous and PL(2) functions). Therefore, the initial inductive assumption made at the beginning of this proof holds at time $t+1$, and we can apply the result inductively to conclude Theorem 1. □

*Proof of Theorem 2.* The proof is an extension of the proof of Theorem 1 to the case where the test function $h$ acts on blocks rather than individual entries. Much of the proof is identical, so we only sketch the argument and highlight the major differences here. We begin with the inductive hypothesis that $\frac{1}{M}\sum_{i=1}^{M}\delta(\boldsymbol{v}_i^{(t)},\boldsymbol{\theta}_i^*) \overset{\mathcal{W}_2}{\to} \Pi_t$, for a known distribution $\Pi_t$ over $\mathbb{R}^b\times\mathbb{R}^b$. Recall here that $M$ denotes the number of blocks/factors of size $b$ (so $M=\frac{d}{b}$).

Then, let $h\colon(\mathbb{R}^b)^3\to\mathbb{R}$ be a test function with $\|\nabla^2 h\|_2\leq C$ and such that either $h$ is bounded or $h(\boldsymbol{u}_i,\boldsymbol{v}_i,\boldsymbol{\theta}_i)=\|\boldsymbol{u}_i\|_2^2$. We consider a similar perturbed optimization problem:

$$P_1(\mu) = \min_{\boldsymbol{u}\in\mathbb{R}^d}\frac{1}{n}\left\|\boldsymbol{y} - \frac{1}{\sqrt{d}}\boldsymbol{X}(\boldsymbol{u}\odot\boldsymbol{v})\right\|_2^2 + \frac{\lambda}{d}\|\boldsymbol{u}\|_2^2 + \frac{\mu}{M}\sum_{i=1}^{M}h(\boldsymbol{u}_i,\boldsymbol{v}_i,\boldsymbol{\theta}_i^*). \tag{22}$$

Again, we consider this for $|\mu|\leq\frac{\lambda}{bC}$, so that the optimization problem is $\frac{\lambda}{d}$ strongly convex in $\boldsymbol{u}$. Noting that the proof of Lemma 1 still holds in this grouped case, we can constrain $P_1(\mu)$ to be over compact sets and apply the CGMT to obtain the auxiliary problem

$$P_2(\mu) = \min_{\boldsymbol{\Delta}\in B_\Delta}\max_{\boldsymbol{q}\in B_q}\frac{2}{\sqrt{n}}\boldsymbol{q}^\top\boldsymbol{\epsilon} - \frac{2}{\sqrt{n}}\|\boldsymbol{q}\|_2\boldsymbol{g}^\top\boldsymbol{\Delta} - \frac{2}{\sqrt{n}}\|\boldsymbol{\Delta}\|_2\boldsymbol{h}^\top\boldsymbol{q} - \|\boldsymbol{q}\|_2^2 + \frac{\lambda}{d}\left\|\frac{\sqrt{d}\boldsymbol{\Delta}+\boldsymbol{\theta}^*}{\boldsymbol{v}}\right\|_2^2 \tag{23}$$

$$+ \frac{\mu}{M}\sum_{i=1}^{M}h(\boldsymbol{u}_i,\boldsymbol{v}_i^{(t)},\boldsymbol{\theta}_i^*). \tag{24}$$

The sequence of "scalarization" steps on $P_2$ is identical to in Theorem 1, until we arrive at

$$P_2(\mu)\overset{d}{=}\max_{0\leq\tau\leq R}\min_{\substack{\boldsymbol{u}\in B_u,\\\sigma\leq\beta\leq\sigma+R}}\frac{\tau\|\boldsymbol{h}\|_2}{\sqrt{n}}\left(\frac{\sigma^2}{\beta} + \frac{\|\boldsymbol{u}\odot\boldsymbol{v}-\boldsymbol{\theta}^*\|_2^2}{\beta d} + \beta\right) - \frac{2\tau}{\sqrt{nd}}\boldsymbol{g}^\top(\boldsymbol{u}\odot\boldsymbol{v}-\boldsymbol{\theta}^*)$$

$$- \tau^2 + \frac{\lambda}{d}\|\boldsymbol{u}\|_2^2 + \frac{\mu}{M}\sum_{i=1}^{M}h(\boldsymbol{u}_i,\boldsymbol{v}_i^{(t)},\boldsymbol{\theta}_i^*). \tag{25}$$

Here, since the sum of the last two terms in the objective function is $\frac{\lambda}{d}$ strongly convex by our choice of $\mu$, the proof of Lemma 2 holds without change and we can consider the unconstrained minimization over $\boldsymbol{u}$. In this case, the minimization is *block-separable* over each of the $M$ factors of $\boldsymbol{u}$, so it can be expressed as

$$\frac{1}{d}\sum_{i=1}^{M}\min_{\boldsymbol{u}_i\in\mathbb{R}^b}\left\{\frac{\tau\xi}{\beta}\|\boldsymbol{u}_i\odot\boldsymbol{v}_i-\boldsymbol{\theta}_i^*\|_2^2 - 2\tau\sqrt{\kappa}\boldsymbol{g}_i^\top(\boldsymbol{u}_i\odot\boldsymbol{v}_i-\boldsymbol{\theta}) + \lambda\|\boldsymbol{u}\|_2^2 + \mu bh(\boldsymbol{u}_i,\boldsymbol{v}_i,\boldsymbol{\theta}_i^*)\right\}$$

$$= \frac{1}{bM}\sum_{i=1}^{M}\min_{\boldsymbol{u}_i\in\mathbb{R}^b}\left\{\frac{\tau\xi}{\beta}\|\boldsymbol{u}_i\odot\boldsymbol{v}_i-\boldsymbol{\theta}_i^*\|_2^2 - 2\tau\sqrt{\kappa}\boldsymbol{g}_i^\top(\boldsymbol{u}_i\odot\boldsymbol{v}_i-\boldsymbol{\theta}) + \lambda\|\boldsymbol{u}\|_2^2 + \mu bh(\boldsymbol{u}_i,\boldsymbol{v}_i,\boldsymbol{\theta}_i^*)\right\}$$

$$:= \frac{1}{bM}\sum_{i=1}^{M}q(\boldsymbol{v}_i,\boldsymbol{\theta}_i^*,\boldsymbol{g}_i),$$

where $\boldsymbol{g}_i \in \mathbb{R}^b$ denotes the $i$th block of $\boldsymbol{g}$ and $q := (\mathbb{R}^b)^3 \to \mathbb{R}$ is defined as a shorthand for the quantity inside the summation.

Next, we consider the asymptotic behavior of $P_2(\mu)$. Here, the only term which is different than in Theorem 1 is the term $\frac{1}{bM} \sum_{i=1}^{M} q(\boldsymbol{v}_i, \boldsymbol{\theta}_i^*, \boldsymbol{g}_i)$. By the same argument as in Lemma 3, we can write $q$ as the Moreau envelope of a convex function and show that

$$\frac{1}{bM} \sum_{i=1}^{M} q(\boldsymbol{v}_i, \boldsymbol{\theta}_i^*, \boldsymbol{g}_i) \xrightarrow{P} \mathbb{E} \frac{1}{b} q(\boldsymbol{V}, \boldsymbol{\Theta}, \boldsymbol{G}),$$

where the expectation is over $(\boldsymbol{V}, \boldsymbol{\Theta}) \sim \Pi_t$ and $\boldsymbol{G} \sim \mathcal{N}(\boldsymbol{0}, \boldsymbol{I}_b)$. After the same uniform convergence argument as in the proof of Theorem 1, we can conclude that for all $\mu \in [-\mu^*, \mu^*]$, $P_1(\mu) \xrightarrow{P} \bar{P}(\mu)$, where

$$\bar{P}(\mu) = \max_{\tau \geq 0} \min_{\beta \geq 0} \frac{\tau \sigma^2}{\beta} + \tau \beta - \tau^2 + \mathbb{E} \frac{1}{b} q(\boldsymbol{V}, \boldsymbol{\Theta}, \boldsymbol{G}).$$

In particular, when $\mu = 0$, the minimization implicit in the definition of $q$ can be solved exactly; this yields exactly the optimization problem in 7. Step 2 of the proof (convergence of test functions of the optimal minimizer) is identical to that of Theorem 1, and for the final step (showing the inductive hypothesis holds at the next iteration), we need to argue that the second moment of $\boldsymbol{v}^{(t+1)} = \psi(\boldsymbol{u}_i^{(t+1)}, \boldsymbol{v}_i)$ converges to its expectation under $\Pi_{t+1}$:

$$\frac{1}{M} \sum_{i=1}^{M} \|\psi(\boldsymbol{u}_i^{(t+1)}, \boldsymbol{v}_i)\|_2^2 \xrightarrow{P} \mathbb{E} \|\psi(\boldsymbol{Q}_{t+1}, \boldsymbol{V})\|_2^2$$

If $\psi$ is bounded and continuous or has PL(2) coordinate projections (as we have assumed), then the above convergence holds based on the Wasserstein-2 convergence of the joint distribution of $(\boldsymbol{u}^{(t+1)}, \boldsymbol{v})$.

$\square$

## B    Technical lemmas

**Proposition 1.** *The function* $g(u, v, \theta) = |uv - \theta|$ *is pseudo-Lipschitz of order 2.*

*Proof.* The result follows from the following series of inequalities:

$$\begin{aligned}
\||uv - \theta| - |u'v' - \theta'\|| &\leq |uv - \theta - (u'v' - \theta')| \\
&\leq |uv - u'v'| + |\theta - \theta'| \\
&\leq |u||v - v'| + |v'||u - u'| + |\theta - \theta'| \\
&\leq (|u| + |v'| + 1)(|u - u'| + |v - v'| + |\theta - \theta'|) \\
&\leq (1 + \|\boldsymbol{x}\|_1 + \|\boldsymbol{x}'\|_1)\|\boldsymbol{x} - \boldsymbol{x}'\|_1 \\
&\leq 3(1 + \|\boldsymbol{x}\|_2 + \|\boldsymbol{x}'\|_2)\|\boldsymbol{x} - \boldsymbol{x}'\|_2,
\end{aligned}$$

where $\boldsymbol{x}, \boldsymbol{x}' \in \mathbb{R}^3$ denote $(u, v, \theta)$ and $(u', v', \theta')$, respectively. $\square$

**Lemma 1.** *Let* $\boldsymbol{\Delta}^*, \boldsymbol{q}^*$ *be the optimal solution to (10). Then, there exists universal constant* $C_1 > 0$ *such that*

$$\lim_{d \to \infty} \mathbb{P}\{\|\boldsymbol{\Delta}^*\|_2 \leq C_1 \|\boldsymbol{v}\|_\infty\} = \lim_{d \to \infty} \mathbb{P}\{\|\boldsymbol{q}^*\|_2 \leq C_1 \|\boldsymbol{v}\|_\infty\} = 1.$$

*Proof.* We proceed via a similar argument to Lemma 2 in [5]. First, consider the original expression for $P_1(\mu)$ from (9) :

$$P_1(\mu) = \min_{\boldsymbol{u} \in \mathbb{R}^d} F(\boldsymbol{u}) + R(\boldsymbol{u}),$$

where $F(\boldsymbol{u}) := \frac{1}{n} \left\| \boldsymbol{y} - \frac{1}{\sqrt{d}} \boldsymbol{X}(\boldsymbol{u} \odot \boldsymbol{v}) \right\|_2^2$ and $R(\boldsymbol{u}) := \frac{\lambda}{d} \|\boldsymbol{u}\|_2^2 + \frac{\mu}{d} \sum_{i=1}^{d} h(u_i, v_i^{(t)}, \theta_i^*)$. Recall here that $R$ is $\frac{\lambda}{d}$-strongly convex for all $\mu \in [-\mu^*, \mu^*]$, and denote the unique optimal minimizer to this

problem as $\boldsymbol{u}^*$. Then, the following chain of inequalities holds, by the optimality of $\boldsymbol{u}^*$ and the non-negativity of $F$.

$$\frac{1}{n}\|\boldsymbol{y}\|_2^2 + R(\boldsymbol{0}) = F(\boldsymbol{0}) + R(\boldsymbol{0}) \geq F(\boldsymbol{u}^*) + R(\boldsymbol{u}^*) \geq R(\boldsymbol{u}^*).$$

Moreover, by the strong convexity of $R$, we have

$$R(\boldsymbol{u}^*) \geq R(\boldsymbol{0}) + \nabla R(\boldsymbol{0})^\top \boldsymbol{u}^* + \frac{\lambda}{2d}\|\boldsymbol{u}^*\|_2^2.$$

Combining the above two series of inequalities, we obtain (recall $\kappa = \frac{d}{n}$)

$$\|\boldsymbol{u}^*\|_2^2 + \frac{2d}{\lambda}\nabla R(\boldsymbol{0})^\top \boldsymbol{u}^* \leq \frac{2\kappa}{\lambda}\|\boldsymbol{y}\|_2^2.$$

After completing the square, this yields

$$\left\|\boldsymbol{u}^* + \frac{d}{\lambda}\nabla R(\boldsymbol{0})\right\|_2^2 \leq \frac{2\kappa}{\lambda}\|\boldsymbol{y}\|_2^2 + \frac{d^2}{\lambda^2}\|\nabla R(\boldsymbol{0})\|_2^2,$$

whence, by the triangle inequality,

$$\|\boldsymbol{u}^*\|_2 \leq \frac{d}{\lambda}\|\nabla R(\boldsymbol{0})\|_2 + \sqrt{\frac{2\kappa}{\lambda}\|\boldsymbol{y}\|_2^2 + \frac{d^2}{\lambda^2}\|\nabla R(\boldsymbol{0})\|_2^2}$$

$$\leq \frac{2d}{\lambda}\|\nabla R(\boldsymbol{0})\|_2 + \sqrt{\frac{2\kappa}{\lambda}}\|\boldsymbol{y}\|_2.$$

Here, standard concentration inequalities for Gaussian random variables (e.g., [32, Theorem 5.2.2, Corollary 7.3.3]) imply that, with probability approaching 1, $\|\boldsymbol{X}\|_2 \lesssim \sqrt{d}$ and $\|\boldsymbol{\epsilon}\|_2 \lesssim \sqrt{d}$. And Assumption 1 implies that $\|\boldsymbol{\theta}^*\|_2 \lesssim \sqrt{d}$ with probability tending to 1. So,

$$\|\boldsymbol{y}\|_2 \leq \frac{\mu}{\sqrt{d}}\|\boldsymbol{X}\|\|\boldsymbol{\theta}^*\|_2 + \|\boldsymbol{\epsilon}\|_2 \lesssim \sqrt{d}$$

with probability approaching 1. Next, we bound $\|\nabla R(\boldsymbol{0})\|_2$. Recalling the definition of $R$,

$$\|\nabla R(\boldsymbol{0})\|_2 = \frac{\mu}{d}\sqrt{\sum_{i=1}^d \left(\frac{\partial}{\partial u}h(u, v_i, \theta_i^*)\Big|_{u=0}\right)^2} = \frac{1}{\sqrt{d}}\sqrt{\frac{1}{d}\sum_{i=1}^d \left(\frac{\partial}{\partial u}h(u, v_i, \theta_i^*)\Big|_{u=0}\right)^2}.$$

Since the function $g(v, \theta) = \frac{\partial}{\partial u}h(u, v, \theta)\Big|_{u=0}$ is Lipschitz (by the fact that $h$ has bounded second derivatives), $g^2$ is pseudo-Lipschitz of order 2. So, the quantity under the square root converges in probability to $\mathbb{E}_{(V,\Theta)\sim\Pi_t}g^2$ by Assumption 1, and, with probability tending to 1, we have $\|\nabla R(\boldsymbol{0})\|_2 \lesssim 1/\sqrt{d}$.

Combining the above bounds on $\|\boldsymbol{y}\|_2$ and $\|\nabla R(\boldsymbol{0})\|_2$, we can conclude that $\|\boldsymbol{u}^*\|_2 \lesssim \sqrt{d}$. The first part of the lemma follows by noting that

$$\|\boldsymbol{\Delta}^*\|_2 = \frac{1}{\sqrt{d}}\|\boldsymbol{u}^* \odot \boldsymbol{v} - \boldsymbol{\theta}^*\|_2 \leq \frac{1}{\sqrt{d}}\|\boldsymbol{u}^* \odot \boldsymbol{v}\|_2 + \frac{1}{\sqrt{d}}\|\boldsymbol{\theta}^*\|_2$$

$$\leq \frac{1}{\sqrt{d}}\|\boldsymbol{v}\|_\infty\|\boldsymbol{u}^*\|_2 + \frac{1}{\sqrt{d}}\|\boldsymbol{\theta}^*\|_2$$

$$\lesssim \|\boldsymbol{v}\|_\infty,$$

where the last inequality holds with probability approaching 1. Lastly, the optimal $\boldsymbol{q}$ for any $\boldsymbol{\Delta}$ has closed-form $\boldsymbol{q} = \frac{1}{\sqrt{n}}\boldsymbol{\epsilon} - \frac{1}{\sqrt{n}}\boldsymbol{X}\boldsymbol{\Delta}$. By the triangle inequality, we then obtain

$$\|\boldsymbol{q}^*\|_2 \leq \frac{1}{\sqrt{n}}\|\boldsymbol{\epsilon}\|_2 + \frac{1}{\sqrt{n}}\|\boldsymbol{X}\|\|\boldsymbol{\Delta}^*\|_2 \lesssim \|\boldsymbol{v}\|_\infty,$$

with the last inequality holding with probability approaching 1, by the concentration of norms of $\boldsymbol{\epsilon}$ and $\boldsymbol{X}$ as discussed above, and the bound on $\|\boldsymbol{\Delta}^*\|_2$. $\qquad\square$

**Lemma 2.** *Consider the following unconstrained minimization problem over $\boldsymbol{u} \in \mathbb{R}^d$:*

$$\min_{\boldsymbol{u} \in \mathbb{R}^d} \max_{0 \leq \tau \leq R} \frac{2\tau \|\boldsymbol{h}\|_2}{\sqrt{n}} \sqrt{\sigma^2 + \frac{1}{d}\|\boldsymbol{u} \odot \boldsymbol{v} - \boldsymbol{\theta}^*\|_2^2} - \frac{2\tau}{\sqrt{nd}}\boldsymbol{g}^\top (\boldsymbol{u} \odot \boldsymbol{v} - \boldsymbol{\theta}^*)$$

$$- \tau^2 + \frac{\lambda}{d}\|\boldsymbol{u}\|_2^2 + \frac{\mu}{d}\sum_{i=1}^{d} h(u_i, v_i^{(t)}, \theta_i^*).$$

*With probability approaching 1, the solution $\boldsymbol{u}^*$ satisfies $\left\|\frac{1}{\sqrt{d}}(\boldsymbol{u}^* \odot \boldsymbol{v} - \boldsymbol{\theta}^*)\right\|_2 \lesssim \|\boldsymbol{v}\|_\infty$.*

*Proof.* First note $\left\|\frac{1}{\sqrt{d}}(\boldsymbol{u}^* \odot \boldsymbol{v} - \boldsymbol{\theta}^*)\right\|_2 \leq \frac{1}{\sqrt{d}}\|\boldsymbol{u}^*\|_2\|\boldsymbol{v}\|_\infty + \frac{1}{\sqrt{d}}\|\boldsymbol{\theta}^*\|_2$, and $\frac{1}{\sqrt{d}}\|\boldsymbol{\theta}^*\|_2$ is bounded by a constant with probability approaching 1 by the assumed $\mathcal{W}_2$ convergence of $\boldsymbol{\theta}^*$ to a fixed limit. Hence, it suffices to show that $\|\boldsymbol{u}^*\|_2 \lesssim \sqrt{d}$ with high probability. We begin by noting that the inner maximization over $\tau$ admits a closed form solution, so the problem becomes

$$\min_{\boldsymbol{u} \in \mathbb{R}^d} \left(\frac{\|\boldsymbol{h}\|_2}{2\sqrt{n}}\sqrt{\sigma^2 + \frac{1}{d}\|\boldsymbol{u} \odot \boldsymbol{v} - \boldsymbol{\theta}^*\|_2^2} - \frac{1}{2\sqrt{nd}}\boldsymbol{g}^\top (\boldsymbol{u} \odot \boldsymbol{v} - \boldsymbol{\theta}^*)\right)_+^2 + \frac{\lambda}{d}\|\boldsymbol{u}\|_2^2 + \frac{\mu}{d}\sum_{i=1}^{d} h(u_i, v_i^{(t)}, \theta_i^*).$$

Now, we can proceed similarly to in the proof of Lemma 1. Let

$$F(\boldsymbol{u}) := \left(\frac{\|\boldsymbol{h}\|_2}{2\sqrt{n}}\sqrt{\sigma^2 + \frac{1}{d}\|\boldsymbol{u} \odot \boldsymbol{v} - \boldsymbol{\theta}^*\|_2^2} - \frac{1}{2\sqrt{nd}}\boldsymbol{g}^\top (\boldsymbol{u} \odot \boldsymbol{v} - \boldsymbol{\theta}^*)\right)_+^2,$$

$$R(\boldsymbol{u}) := \frac{\lambda}{d}\|\boldsymbol{u}\|_2^2 + \frac{\mu}{d}\sum_{i=1}^{d} h(u_i, v_i^{(t)}, \theta_i^*).$$

Then, noting $F$ is always non-negative and $R$ is $\frac{\lambda}{d}$ strongly-convex, we use the same argument as in Lemma 1 to obtain the inequality

$$\|\boldsymbol{u}^*\|_2^2 + \frac{2d}{\lambda}\nabla R(\boldsymbol{0})^\top \boldsymbol{u}^* \leq \frac{2d}{\lambda}F(\boldsymbol{0}).$$

After completing the squares, we obtain

$$\left\|\boldsymbol{u}^* + \frac{d}{\lambda}\nabla R(\boldsymbol{0})\right\|_2^2 \leq \frac{2d}{\lambda}F(\boldsymbol{0}) + \frac{d^2}{\lambda^2}\|\nabla R(\boldsymbol{0})\|_2^2,$$

so we can conclude

$$\|\boldsymbol{u}^*\|_2 \leq \frac{d}{\lambda}\|\nabla R(\boldsymbol{0})\|_2 + \sqrt{\frac{2d}{\lambda}F(\boldsymbol{0}) + \frac{d^2}{\lambda^2}\|\nabla R(\boldsymbol{0})\|_2^2} \leq \frac{2d}{\lambda}\|\nabla R(\boldsymbol{0})\|_2 + \sqrt{\frac{2d}{\lambda}F(\boldsymbol{0})}.$$

As shown in Lemma 1, $\|\nabla R(\boldsymbol{0})\|_2 \lesssim \frac{1}{\sqrt{d}}$ with probability approaching 1. It remains to show that $\sqrt{F(\boldsymbol{0})} \lesssim 1$ with probability approaching 1. To see this, observe that

$$\sqrt{F(\boldsymbol{0})} = \left(\frac{\|\boldsymbol{h}\|_2}{2\sqrt{n}}\sqrt{\sigma^2 + \frac{1}{d}\|\boldsymbol{\theta}^*\|_2^2} - \frac{1}{2\sqrt{nd}}\boldsymbol{g}^\top \boldsymbol{\theta}^*\right)_+$$

$$\leq \left|\frac{\|\boldsymbol{h}\|_2}{2\sqrt{n}}\sqrt{\sigma^2 + \frac{1}{d}\|\boldsymbol{\theta}^*\|_2^2} - \frac{1}{2\sqrt{nd}}\boldsymbol{g}^\top \boldsymbol{\theta}^*\right|$$

$$\leq \frac{\|\boldsymbol{h}\|_2}{2\sqrt{n}}\sqrt{\sigma^2 + \frac{1}{d}\|\boldsymbol{\theta}^*\|_2^2} + \frac{1}{2\sqrt{nd}}\|\boldsymbol{g}\|_2\|\boldsymbol{\theta}^*\|_2,$$

where the last line uses the triangle and Cauchy-Schwarz inequalities. By concentration of the norm for Gaussian vectors, there exists a universal constant $c$ such that $\frac{\|\boldsymbol{h}\|_2}{\sqrt{n}} \leq c$ and $\frac{\|\boldsymbol{g}\|_2}{\sqrt{n}} \leq c$ with probability approaching 1. Moreover, by Assumption 1, $\frac{\|\boldsymbol{\theta}^*\|_2}{\sqrt{d}} \leq c$ with probability approaching 1. Hence, $\sqrt{F(\boldsymbol{0})} \lesssim 1$ with probability approaching 1. Substituting this into the bound for $\|\boldsymbol{u}^*\|_2$ from above completes the proof. $\square$

**Lemma 3.** *Under Assumption 1, the function $f_d(\tau, \beta)$ (Eq. 19) converges pointwise in probability to $f(\tau, \beta)$ (Eq. 20) as $d \to \infty$.*

*Proof.* We consider the limit of each term in $f_d$ separately. The limit of the terms $\frac{\tau\sigma^2\xi}{\beta}$ and $\tau\beta\xi$ is found by noting that $\xi \to 1$ in probability by Gaussian Lipschitz concentration.

The first summation term simplifies as follows:

$$\frac{1}{d}\sum_{i=1}^{d}\left[\frac{\tau\xi\theta_i^{*2}}{\beta} - \frac{\tau}{\beta\xi}\left(\beta g_i\sqrt{\kappa} - \xi\theta_i^*\right)^2\right] = \frac{1}{d}\sum_{i=1}^{d}\left[-\frac{\tau}{\beta\xi}\left(\beta^2 g_i^2\kappa - 2\beta g_i\sqrt{\kappa}\xi\theta_i^*\right)\right]$$

$$= -\frac{\tau}{\beta\xi}\frac{1}{d}\sum_{i=1}^{d}\left[\beta^2 g_i^2\kappa - 2\beta g_i\sqrt{\kappa}\xi\theta_i^*\right] \xrightarrow{P} -\tau\beta\kappa,$$

where the last line follows from the weak law of large numbers since the $g_i$ are i.i.d. standard Gaussian variables.

For the last term, after again using the fact that $\xi \xrightarrow{P} 1$, we need to consider

$$\frac{1}{d}\sum_{i=1}^{d}\left[\mathcal{M}_{\lambda(\cdot)^2+\mu h(\cdot, v_i^{(t)}, \theta_i^*)}\left(\frac{\theta_i^* - \beta g_i\sqrt{\kappa}}{v_i^{(t)}}; \frac{\beta}{2\tau v_i^{(t)2}}\right)\right] =: \frac{1}{d}\sum_{i=1}^{d}q(v_i, \theta_i^*, g_i)$$

To show convergence in probability of this term, first fix $\delta > 0$. Then, we want to show

$$\mathbb{P}\left[\left|\frac{1}{d}\sum_{i=1}^{d}q(v_i, \theta_i^*, g_i) - \mathbb{E}\,q(V, \Theta, G)\right| > \delta\right] \to 0,$$

where the expectation is over $(V, \Theta) \sim \Pi_t$ and $G \sim \mathcal{N}(0, 1)$. It suffices to show the following two statements:

$$\mathbb{P}\left[\left|\frac{1}{d}\sum_{i=1}^{d}q(v_i, \theta_i^*, g_i) - \mathbb{E}_{\boldsymbol{g}}\frac{1}{d}\sum_{i=1}^{d}q(v_i, \theta_i^*, g_i)\right| > \frac{\delta}{2}\right] \to 0, \tag{26}$$

$$\mathbb{P}\left[\left|\mathbb{E}_{\boldsymbol{g}}\frac{1}{d}\sum_{i=1}^{d}q(v_i, \theta_i^*, g_i) - \mathbb{E}\,q(V, \Theta, G)\right| > \frac{\delta}{2}\right] \to 0. \tag{27}$$

To show (26), we rely on a concentration inequality for the Moreau envelope of a Gaussian vector plus a bounded vector from [5]. First note that

$$\frac{1}{d}\sum_{i=1}^{d}q(v_i, \theta_i^*, g_i) = \frac{1}{d}\min_{\boldsymbol{u}\in\mathbb{R}^d}\left\{\lambda\|\boldsymbol{u}\|_2^2 + \mu\sum_{i=1}^{d}h(u_i, v_i, \theta_i^*) + \frac{\tau}{\beta}\|\boldsymbol{u}\odot\boldsymbol{v} - \boldsymbol{\theta}^* + \beta\sqrt{\kappa}\boldsymbol{g}\|_2^2\right\}$$

$$= \frac{1}{d}\min_{\boldsymbol{u}\in\mathbb{R}^d}\left\{\lambda\|\boldsymbol{u}\|_2^2 + \mu\sum_{i=1}^{d}h(u_i, v_i, \theta_i^*) + \tau\beta\kappa\left\|\frac{\boldsymbol{u}\odot\boldsymbol{v}}{\beta\sqrt{\kappa}} - \frac{\boldsymbol{\theta}^*}{\beta\sqrt{\kappa}} + \boldsymbol{g}\right\|_2^2\right\}$$

$$= \frac{1}{d}\min_{\boldsymbol{\theta}\in\mathbb{R}^d}\left\{\lambda\left\|\frac{\beta\sqrt{\kappa}\boldsymbol{\theta}}{\boldsymbol{v}}\right\|_2^2 + \mu\sum_{i=1}^{d}h\left(\frac{\beta\sqrt{\kappa}\theta_i}{v_i}, v_i, \theta_i^*\right) + \tau\beta\kappa\left\|\boldsymbol{\theta} - \frac{\boldsymbol{\theta}^*}{\beta\sqrt{\kappa}} + \boldsymbol{g}\right\|_2^2\right\}$$

$$= \frac{1}{d}\mathcal{M}_\ell\left(\frac{\boldsymbol{\theta}^*}{\beta\sqrt{\kappa}} - \boldsymbol{g}; \frac{1}{2\tau\beta\kappa}\right),$$

where the second to last line follows from the change of variable $\boldsymbol{\theta} = \frac{\boldsymbol{u}\odot\boldsymbol{v}}{\beta\sqrt{\kappa}}$, and $\ell(\boldsymbol{\theta}) := \lambda\left\|\frac{\beta\sqrt{\kappa}\boldsymbol{\theta}}{\boldsymbol{v}}\right\|_2^2 + \mu\sum_{i=1}^{d}h\left(\frac{\beta\sqrt{\kappa}\theta_i}{v_i}, v_i, \theta_i^*\right)$. Here, for fixed $\boldsymbol{v}$, $\ell$ is a proper convex function of $\boldsymbol{\theta}$. Moreover, by Assumption 1, $\frac{\boldsymbol{\theta}^*}{\beta\sqrt{\kappa}}$ has norm of order $\sqrt{d}$ with high probability. Hence, by [5, Lemma 8], this quantity concentrates around its expectation (with respect to $\boldsymbol{g}$), and we can conclude that there is some $c > 0$ such that

$$\mathbb{P}\left[\left|\frac{1}{d}\sum_{i=1}^{d}q(v_i, \theta_i^*, g_i) - \mathbb{E}_{\boldsymbol{g}}\frac{1}{d}\sum_{i=1}^{d}q(v_i, \theta_i^*, g_i)\right| > \frac{\delta}{2}\right] \le \frac{c\tau^2\beta^2\kappa^2}{d\delta^2} \to 0.$$

To show (27), note that

$$\mathbb{E}_{\boldsymbol{g}} \frac{1}{d} \sum_{i=1}^{d} q(v_i, \theta_i^*, g_i) = \frac{1}{d} \sum_{i=1}^{d} \mathbb{E}_G \, q(v_i, \theta_i^*, G).$$

Observe that this quantity is an expectation with respect to the joint empirical distribution of $(\boldsymbol{v}, \boldsymbol{\theta}^*)$. By the assumption of $\mathcal{W}_2$ convergence of the empirical distribution of $(\boldsymbol{v}, \boldsymbol{\theta}^*)$ (Assumption 1), if we can show that mapping $(v, \theta) \to \mathbb{E}_G \, q(v, \theta, G)$ is bounded, then we can conclude that the above quantity converges in probability to $\mathbb{E} \, q(V, \Theta, G)$, with $(V, \Theta) \sim \Pi_t$. To see this, recall that for all $G \in \mathbb{R}$, $q$ is bounded below as

$$q(v, \theta, G) \geq \min_u \lambda u^2 + \mu h(u, v, \theta),$$

which is always bounded below since $h$ is bounded (or, in the case $h = u^2$, the lower bound is zero). Next, for a given $G$, we can bound $q$ above as

$$q(v, \theta, G) \leq \mu h(0, v, \theta) + \frac{\tau}{\beta}(\theta - \beta \sqrt{\kappa} G)^2.$$

Hence,

$$\mathbb{E}_G \, q(v, \theta, G) \leq \mu h(0, v, \theta) + \frac{\tau}{\beta}\theta^2 + \tau \beta \kappa < C$$

for some universal constant $C > 0$, since $\theta$ is bounded by assumption and $h(0, v, \theta)$ is either bounded above or equal to 0 (in the case where $h = u^2$). Combining (26) and (27) yields the desired result. $\quad\square$

## C   Solving the min-max problem

Below, we show that the max-min problems (5) and (7) have easily computable solutions. Note it suffices to consider the grouped case (7), since we can apply it with $b = 1$ to recover the ungrouped case. The following derivation closely follows the analysis of the scalar max-min problem in [8], who study a similar scalar problem (albeit in the case of $\lambda = 0$, which we do not consider).

First recall that, as shown in the proof of Theorem 1, there exists a unique saddle point, due to the strong convexity in $\tau$ and strict convexity in $\beta$. Now, taking derivatives with respect to $\tau$ and $\beta$, we obtain the following saddle point conditions:

$$0 = \frac{\sigma^2}{\beta} + \beta(1 - \kappa) - 2\tau + \beta\lambda^2 \, \mathbb{E}\left[\frac{1}{b}\sum_{i=1}^{b} \frac{\Theta_i^2 + \beta^2 \kappa}{(\tau V_i^2 + \beta\lambda)^2}\right], \tag{28}$$

$$0 = -\frac{\tau\sigma^2}{\beta^2} + \tau(1 - \kappa) + \tau\lambda \, \mathbb{E}\left[\frac{1}{b}\sum_{i=1}^{b} \frac{1}{\tau V_i^2 + \beta\lambda}\right] - \tau\lambda^2 \, \mathbb{E}\left[\frac{1}{b}\sum_{i=1}^{b} \frac{\Theta_i^2 + \beta^2 \kappa}{(\tau V_i^2 + \beta\lambda)^2}\right]. \tag{29}$$

Solving each of these equations for the quantity $\beta^2 \lambda^2 \, \mathbb{E}\left[\frac{1}{b}\sum_{i=1}^{b} \frac{\Theta_i^2 + \beta^2 \kappa}{(\tau V_i^2 + \beta\lambda)^2}\right]$ and then equating the two, we arrive at

$$2\tau\beta - \beta^2(1 - \kappa) - \sigma^2 = \beta^2(1 - \kappa) - \sigma^2 + 2\lambda\beta^3\kappa \, \mathbb{E}\left[\frac{1}{b}\sum_{i=1}^{b} \frac{1}{\tau V_i^2 + \beta\lambda}\right],$$

which implies

$$\tau = \beta(1 - \kappa) + \lambda\beta^2\kappa \, \mathbb{E}\left[\frac{1}{b}\sum_{i=1}^{b} \frac{1}{\tau V_i^2 + \beta\lambda}\right].$$

Defining the auxiliary variable $\gamma = \tau/\beta$, this yields the fixed point equation

$$\gamma = 1 - \kappa + \lambda\kappa \, \mathbb{E}\left[\frac{1}{b}\sum_{i=1}^{b} \frac{1}{\gamma V_i^2 + \lambda}\right]. \tag{30}$$

Substituting this $\gamma$ back into the first optimality condition in (28), we can express the optimal $\beta$ in closed-form, in terms of $\gamma$:

$$\beta = \sqrt{\frac{\sigma^2 + \lambda^2 \, \mathbb{E}\left[\frac{1}{b} \sum_{i=1}^{b} \frac{\Theta_i^2}{(\gamma V_i^2 + \lambda)^2}\right]}{2\gamma + \kappa - 1 - \lambda^2 \kappa \, \mathbb{E}\left[\frac{1}{b} \sum_{i=1}^{b} \frac{1}{(\gamma V_i^2 + \lambda)^2}\right]}}.$$

Finally, the optimal $\tau$ can be found simply as $\tau = \gamma \beta$.

This yields a simple recipe for solving the min-max problem. First, compute the positive solution $\hat{\gamma}$ to the fixed point equation (30) (this can be found easily using standard numerical solvers). Then, $(\hat{\beta}, \hat{\tau})$ are both given in closed-form as functions of $\hat{\gamma}$ (where the required expectations can all be approximated via Monte Carlo simulation).

# D  Further simulations

In this section, we demonstrate that our asymptotic predictions can provide accurate estimates of the test error, even when some of our technical assumptions are not satisfied.

First, we compare the two "heavier" weightings considered in Section 3.1, $\psi(u, v) = \tanh|uv|$ and $\psi(u, v) = \tanh u^2$, to the same weightings without the bounded tanh activation: $\psi(u, v) = |uv|$ and $\psi(u, v) = u^2$. We note that the reweighting choice $|uv|$ is considered in [21, 28] as a limit as $p \to 0$ of the classical IRLS update for $\ell_p$ minimization. In Figure 3a, we consider the same sparse regression as in Section 3.1, i.e., with $n = 250, d = 2000, \sigma = 0.1, \theta_i^* \overset{\text{i.i.d.}}{\sim} \text{Bernoulli}(0.01)$ and $\lambda$ chosen to minimize the predicted asymptotic loss.

For each choice of $\psi$, we apply the theoretical predictions of Theorem 1, even if $\psi$ violates Assumption 2. We find that our predictions remain accurate for all these choices of $\psi$. The choice $\tanh|uv|$ performs almost identically without the tanh activation. Interestingly, the choice $\psi = \tanh u^2$ outperforms the variant without the tanh and has a more regular decay of the test loss.

In Figure 3b, we apply Theorem 1 to predict the asymptotic squared test loss: $\frac{1}{d}\|\boldsymbol{u} \odot \boldsymbol{v} - \boldsymbol{\theta}^*\|_2^2$ at each iteration. While this function is not $PL(2)$, as required by the theorem, the asymptotic predictions still align well with simulations. Extending our technical results to hold formally in such scenarios is an interesting direction for future work.

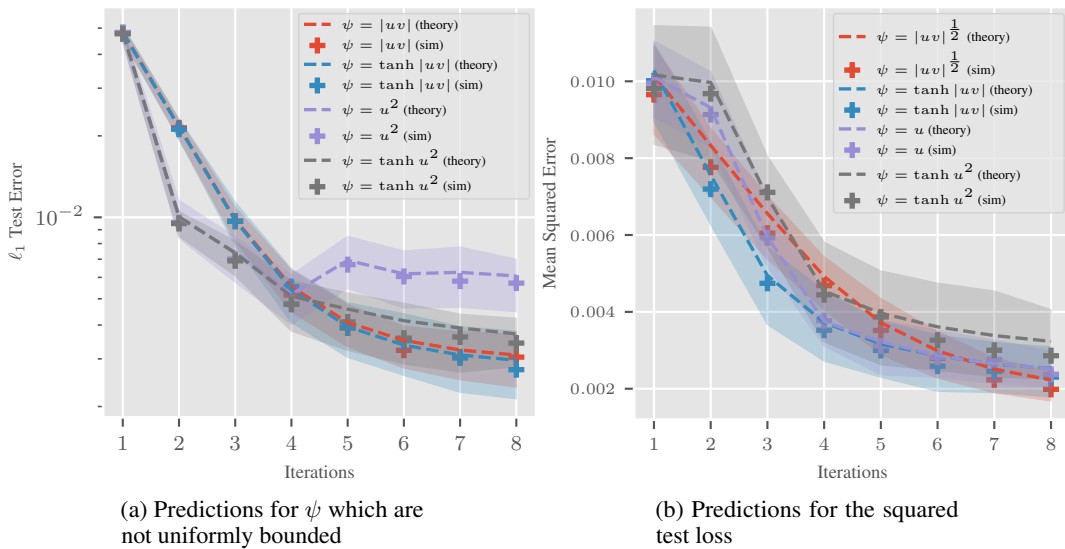

(a) Predictions for $\psi$ which are not uniformly bounded

(b) Predictions for the squared test loss

Figure 3: Here, we fix $n = 250, d = 2000, \sigma = 0.1, \theta_i^* \overset{\text{i.i.d.}}{\sim} \text{Bernoulli}(0.01)$ and select $\lambda$ to minimize the predicted asymptotic loss. Plus marks denote the median over 100 trials, and the shaded region indicates the interquartile range. Left: Predictions and simulations for weighting functions which are not uniformly bounded. Right: Predictions and simulations for the squared error $\frac{1}{d}\|\boldsymbol{u} \odot \boldsymbol{v} - \boldsymbol{\theta}^*\|_2^2$.

