# OpenReview forum: "Precise asymptotics of reweighted least-squares algorithms for linear diagonal networks"
_NeurIPS.cc/2024/Conference — NeurIPS 2024 poster_

### Official Review · Reviewer_kkyZ · 2024-07-02

**Soundness:** 4
**Presentation:** 4
**Contribution:** 3
**Rating:** 7
**Confidence:** 4

**Summary:**

This paper studies the trajectory of a certain iterative scheme in the high-dimensional limit.
In each step, there're two update steps on a fresh batch: 1) regularized least squares; 2) element-wise nonlinear transformation.
This type of iteration is motivated and encompasses several interesting iterations such as IRLS, AGOP, etc.

The result is a precise characterization of the empirical distribution of the iterates at any finite time in the proportional limit.
This allows one to compute many interesting summary statistics.

**Strengths:**

Though the iteration studied in this paper may look exotic at the first glance, it is well motivated (see the first two pages).
The proof uses, unsurprisingly, CGMT since the problem is given in a form that's ready for it.
That said, working out the details is not a trivial task.
I went through the whole argument in the appendix and got the impression that everything is treated in a fairly rigorous way.
In particular, all technicalities associated with CGMT such as justifying the exchange of min / max and proving uniform convergence have been properly addressed.
In doing so, the authors have invoked various tricks / auxiliary results commonly used in this literature.

**Weaknesses:**

As the authors have already discussed, one obvious drawback of CGMT is that it needs sample splitting (which may make the iteration under consideration incompatible with its motivating special cases).
I actually think that CGMT has the potential to handle repeated batches, or even full batch, though the resulting state evolution is likely to be much less interpretable.
So I don't really object to the sample splitting assumption.

Also, CGMT is capable of offering non-asymptotic guarantees.
I wonder if the authors are interested in doing so (no pressure).

**Questions:**

Minor comments:

1. In the equation between line 66/67, the fresh dataset $\boldsymbol{y}^{(t)}, \boldsymbol{X}^{(t)}$ hasn't been defined (correct me otherwise).

1. Line 49, typo: is --> are

1. Line 164, typo: $t = 0, 1, \dots, T$

1. Equation between line 165/166, why not add superscript $(t)$ to $\boldsymbol{\epsilon}$?
After all, the noise is also fresh and is not shared by other batches.

1. Regarding boundedness of $\boldsymbol{\theta}^*$, as mentioned, it's only used in Lemma 3.
Over there, I wonder if it's possible to remove this by truncating $\boldsymbol{\theta}^*$ entry-wise at a large constant $K$, doing the proof (in particular the argument in line 682-692), then sending $K\to\infty$.
(To clarify, this is a minor assumption and removing it requires technical work that may not worth it.
So I don't mind if the authors leave it as is.)

1. Line 246-249, the authors meant to discuss a second difference from prior works.
What's done in this paper is discussed but what's done in prior work is not.
So the comparison is unclear.

1. There are two notation sections that are almost identical. I suggest remove one of them.

1. Line 462, what is a "proper function"? Do you mean a "proper convex function"?

1. Line 466, "restate": please put a reference for CGMT.

1. Equation between line 577/578, $V$ should be $V_t$.

1. Equation between line 579/580, $v_i$ should be $v_i^{(t)}$, $V$ should be $V_t$.

1. Line 580, typo: "satisfies".

---

> ### Author Rebuttal · Authors · 2024-08-06
>
> Thank you for the thorough review of our technical results and constructive feedback. We agree that obtaining non-asymptotic guarantees for this problem would be a very interesting technical challenge for future work. Here, we rely explicitly on a distributional characterization of the iterates, rather than obtaining a state evolution over deterministic scalars. This makes obtaining non-asymptotic guarantees more challenging (although likely possible) after the first 1-2 iterations.
>
> We address your individual questions below. Thanks for catching some typos and the missing CGMT reference - we will fix all of these in our revision. Regarding a few individual questions:
>
> - Boundedness of $\theta^*$: Thanks for this suggestion! Yes, we do believe this kind of truncation argument should allow us to weaken the boundedness requirement on $\theta^*$ to some weaker regularity condition on the distribution. However, we seem to run into some issues when trying to apply a truncation result recursively for each iteration (since we end up needing convergence of the empirical distribution of $(v, \theta^*)$ in a higher order Wasserstein metric). We will continue working on this and update the manuscript if we find a way around it!
> - Line 246-249: Thanks for the catch. The prior work [CLOT21] assumes convergence of the initialization in $W_4$ and proves convergence of the estimator in $W_3$, which would not work for a recursive application of the result. We will add this in the revision.
>
> References:
>
> [CLOT21] Chang et al., “Provable benefits of overparameterization in model compression: From double descent to pruning neural networks,” AAAI 2021.

---

> > ### Comment · Reviewer_kkyZ · 2024-08-08
> >
> > I thank the authors for their comments (including the global one).
> > I took a brief look at them and keep my evaluation unchanged.

---

### Official Review · Reviewer_cDEg · 2024-07-07

**Soundness:** 3
**Presentation:** 3
**Contribution:** 3
**Rating:** 6
**Confidence:** 4

**Summary:**

The paper considers a general algorithm of the form shown in Lines 66-67, as it includes several interesting methods as special cases.

The main result is Theorem 1, which proves asymptotic convergence in probability of any function $g\in PL(2)$ to the corresponding expectation. One important special case of this result is the convergence of the test errors of the iterates.

Synthetic experiments show that the errors predicted by Theorem 1 align well with the actual performance of the algorithms.

**Strengths:**

- The paper is well-written, easy to follow, and well-motivated.
- One interesting contribution is that the paper understands a few different lines of research and is able to unify them using a general algorithm, where the reweighting function can be chosen to implement a specific type of method.
- Theorem 1 is a powerful result as it holds for any function in the specific class $PL(2)$ or for any continuous and bounded function.
- The proofs do not seem to be trivial to me. While I am not familiar with the proof techniques, the paper illustrates the technical points in the paragraph of Line 240, which makes the high-level proof idea clear and provides cues for interested readers to go deeper into the technical details.

**Weaknesses:**

- The first weakness is the assumption that each batch of data is independent. Personally I feel that this is a very strict assumption as I can not imagine a situation where it is satisfied. For example, almost all optimization algorithms (say SGD) would also reuse the previously seen data. Perhaps it is difficult to remove this assumption, but I think more discussions about it could be beneficial. Specifically:
   - Could the authors please provide experiments similar to Figure 1 and allow the algorithm to reuse the data? (it is not very clear to me whether Figure 1 takes fresh batches or not.)
   - Could authors please comment on the technical hurdles if the independence assumption is not available?
   - Prior analysis on IRLS does not have this assumption, see for example: Theorem 2 of "On the Convergence of IRLS and Its Variants in Outlier-Robust Estimation" and in particular Figure 2a shows the error prediction; Theorem 1 of "Globally-convergent Iteratively Reweighted Least Squares for Robust Regression Problems"; Theorem 3.2 of "Iteratively Reweighted Least Squares for Basis
Pursuit with Global Linear Convergence Rate". Would their analysis be useful in any way to remove the independence assumption?

- The second weakness is that the experimental evaluations in their current form are not very deep. For example:
   - Is the error bound prediction accurate for $\ell_2$ squared test loss? The paper commented that $\ell_2$ squared loss does not belong to $PL(2)$, but having this experiment is important as $\ell_2$ squared losses are more commonly seen in my opinion and this experiment can verify whether the assumption of $PL(2)$ is too strong.
   - It seems a little bit weird that alternating minimization exhibits non-monotonic errors in Figure 1b (purple). Is that because these are test errors, or because the $\ell_1$ loss is a bit different from the training loss (objective function)? How about $\ell_2$ losses or the prediction (residual) errors on test (training) data? All these can be easily verified by experiments, and having them would give a better understanding of the behavior of alternating minimization, and it would also make the analysis more complete and convincing in my humble opinion.

**Questions:**

- Is it necessary to define pseudo-Lipschitz of order $k$? It seems that for most of the cases, $k$ is simply chosen to be $2$.
- Lines 185-186: What is the precise relation between $\alpha$ and $p$? It is not very clear why we need two different symbols.

**Limitations:**

See above.

---

> ### Author Rebuttal · Authors · 2024-08-06
>
> Thank you for the detailed review and constructive feedback. We address your individual questions and concerns below.
>
> *(1) “Could the authors please provide experiments similar to Figure 1 and allow the algorithm to reuse the data? (it is not very clear to me whether Figure 1 takes fresh batches or not.)”*
>
> We obtain all figures in the paper by performing the iteration in equation 4, which has a fresh batch at each step. We will be sure to specify this explicitly in each figure caption. In general, we do not expect the same exact behavior in the “full-batch” setting (indeed, the total amount of data observed is smaller by a factor of T), but a similar behavior holds when samples are randomly chosen at each iteration (with repetition allowed). Please see part (a) of our global response for more discussion and supplemental simulations for this point.
>
> *(2) “Could authors please comment on the technical hurdles if the independence assumption is not available?”*
>
> Please see part (a) of our global response for discussion of this point.
>
> *(3) “Prior analysis on IRLS does not have this assumption…”*
>
> We agree that prior non-asymptotic convergence results for IRLS do not have this assumption. However, we emphasize that the type of result we achieve is substantially different in flavor and relies on a very different set of technical tools. Unlike these works, we aim to provide an exact asymptotic characterization of the distribution of the iterates in the high-dimensional limit. While the tools required for this necessitate stronger assumptions on the data (Gaussian) and algorithm (online/sample-split setting), the resultant guarantees are much stronger and can apply to a wider range of problem settings and algorithms.
>
> *(4) “Is the error bound prediction accurate for squared test loss?...”*
>
> We have included simulations for the squared loss in Appendix D. Even though the squared loss is not PL(2), we do find that the error prediction from Theorem 1 seems to still be accurate in this case. So, even though our result doesn’t formally hold for squared loss, we still believe our theorem can still be useful for trajectory predictions in this case.
>
> *(5) “It seems a little bit weird that alternating minimization exhibits non-monotonic errors in Figure 1b…”*
>
> We were also surprised by this behavior! We find that, rather than depending on the loss used ($\ell_1$ or $\ell_2$), this seems to depend on the noise level, with the non-monotonicity appearing more prominently in the low-noise regime (Fig. 1b). We agree that this deserves more mention in the main paper, so we will be sure to mention this in Section 3.1.
>
> *(6) “Is it necessary to define pseudo-Lipschitz of order k?”*
>
> It is true that we only use the choice k=2 in our analysis – we will simplify the definition accordingly.
>
> *(7) “Lines 185-186: What is the precise relation between $\alpha$ and $p$?*
>
> Thanks for the question – this reweighting function is equivalent to the one used by the IRLS-p algorithm from [MF12] with the choice $p=2-4\alpha$. The analysis in [DDFG10] also makes an explicit connection between IRLS updates of this form and $\ell_p$-minimization for $0<p\leq1$.
>
> References:
>
> [MF12] Mohan and Fazel, "Iterative reweighted algorithms for matrix rank minimization," JMLR 2012.
>
> [DDFG10] Daubechies et al., "Iteratively reweighted least squares minimization for sparse recovery," *Communications on Pure and Applied Mathematics,* 2010.

---

> ### Comment · Reviewer_cDEg · 2024-08-11
> **Reply**
>
> Dear authors, thank you for your reply to my comments.
>
> I have taken a look at the rebuttal and also the comments from other reviewers.
>
> I would like to keep my current, positive score, given the quality of the paper.

---

### Official Review · Reviewer_yJHL · 2024-07-10

**Soundness:** 4
**Presentation:** 3
**Contribution:** 3
**Rating:** 5
**Confidence:** 4

**Summary:**

In this paper, the authors propose theoretical analysis of the high-dimensional dynamics of the reweighted least-squares methods in the context of "linear diagonal networks".
The general algorithm is given in Equation (4) and includes alternating minimization (AM), reparameterized IRLS, and linear recursive feature machines (lin-RFM) as special cases with different choices of reweighting function, see Table 1.
In the high-dimensional regime where the observation dimension $d$ and sample size $n$ are both large and comparable, the authors provide, in Theorem 1, distributional characterization of the algorithm behavior for a fixed number of iteration $T$ as $n,d \to \infty$ at the same pace.
The result is then extended the multi-variate setting in Theorem 2.
Simulations results are provided to support the proposed theoretical analyses.

**Strengths:**

The paper is in general in good shape.
It is well written and easy to follow.
The proposed analysis is based on Convex Gaussian Min-Max Theorem and improves previous efforts.

**Weaknesses:**

It seems that the problem under study is of interest, but it would be great if the author would better motivate the study and setting, e.g., by providing some take-home messages.
I have some detailed comments below

**Questions:**

* line 77 contribution: I suggest that the authors could consider adding pointers to precise theoretical/simulation results when talking about contributions. For example, referring to Theorem 1 for the first contribution.
* line 91, perhaps add a paragraph on the related technique tool of Convex Gaussian Min-Max Theorem beyond [7], what is the technical challenge here, and whether the obtained technical results are of more general interest?
* line 172-175: the assumption of $nT > d$ is an artifact of the proof, or very intrinsic to the obtained theoretical results? I somehow feel this is a bit unrealistic, or at least, a regime that is not of great interest to study. Could the authors comment on this?
Also, this should be stated explicitly in the theorem.
* the authors mentioned a few time the keyword "feature learning", but after reading the paper I did not find any in-depth discussion on this. Perhaps having an additional remark to make the connection?

minor:
* line 197: of the algorithm in (4) or in Equation (4).

**Limitations:**

I do not see any potential negative social impact of this work.

---

> ### Author Rebuttal · Authors · 2024-08-06
>
> Thank you for the detailed review and constructive feedback. We address your individual questions and concerns below.
>
> *(1) “It seems that the problem under study is of interest, but it would be great if the author would better motivate the study and setting, e.g., by providing some take-home messages.”*
>
> Thanks for the feedback - please see part (b) of our global response for the main take-home messages we hope to convey. In particular, we would like to emphasize the flexibility of our framework in characterizing the test error of several previously proposed algorithms which have been shown to perform well for sparse recovery tasks or to connect with feature learning in linear neural nets. Our results also provide theoretical support for the use of “weight sharing” to learn signals with additional group-sparse structure. We plan to emphasize these points more explicitly in the final discussion section.
>
> *(2) “line 77 contribution: I suggest that the authors could consider adding pointers to precise theoretical/simulation results when talking about contributions. For example, referring to Theorem 1 for the first contribution.”*
>
> Thanks for the suggestion - in the revision, we will include the appropriate references.
>
> *(3) “line 91, perhaps add a paragraph on the related technique tool of Convex Gaussian Min-Max Theorem beyond [7], what is the technical challenge here, and whether the obtained technical results are of more general interest?”*
>
> While the CGMT has been used extensively in recent literature to analyze estimation problems, a few new technical challenges arise when applying the result to an entire optimization trajectory, since each step relies on the previous iterate. The proof in [CPT23] relies on the fact that the “auxiliary optimization” obtained via the CGMT can be written in terms of a scalar functional of the previous iterate. However, the LDNN parameterization does not fall into the class of problems covered by this analysis technique. To deal with this, we instead obtain a distributional characterization of the iterates at each step. We do believe the proof technique for obtaining Wasserstein convergence guarantees could extend to studying other iterative algorithms in the sample-split setting. We state a few of these points in the last paragraph of the related work (Lines 141-147), but we will plan on adding a short description of the proof aspects we use which may be of more general interest.
>
> *(4) “line 172-175: the assumption of nT > d is an artifact of the proof, or very intrinsic to the obtained theoretical results? I somehow feel this is a bit unrealistic, or at least, a regime that is not of great interest to study. Could the authors comment on this? Also, this should be stated explicitly in the theorem.”*
>
> The setting that we emphasize here is actually the scenario where $nT < d$, so the total amount of observed data is smaller than the dimensionality. In such settings, effective algorithms must be able to take advantage of the signal sparsity to achieve good performance. While we focus on this more interesting regime, our technical results do not depend on this assumption and in fact hold for any constant value of $\kappa = d/n$.
>
> *(5) “the authors mentioned a few time the keyword "feature learning", but after reading the paper I did not find any in-depth discussion on this. Perhaps having an additional remark to make the connection?”*
>
> Thanks for the feedback - please see part (b) of our global response about this point. We will be sure to add an additional remark about this in revision.
>
> *(6) “line 197: of the algorithm in (4) or in Equation (4).”*
>
> Thanks - we will fix this in the revision.
>
> References:
>
> [CPT23] Chandrasekher et al. “Sharp global convergence guarantees for iterative nonconvex optimization with random data,” The Annals of Statistics, 2023.

---

### Author Rebuttal · Authors · 2024-08-06

We thank all the reviewers for their time and helpful feedback. We will carefully consider and incorporate all the suggestions into the next revision of this paper.

We address here some of the main comments:

**(a) Sample-splitting assumption:** We agree that the sample-splitting assumption deviates from the existing literature on IRLS-style algorithms and linear diagonal neural networks. The primary technical reason for this assumption lies in application of the Convex Gaussian Min-Max Theorem after the first step of the optimization algorithm. In particular, the CGMT (cf. Theorem 3) requires an objective function where the only dependence on the Gaussian random matrix is in a bilinear term. If the previous iterate also depends on the same data matrix, this condition is violated. However, we believe that this setting is still of practical interest, e.g., in online/streaming data settings where data is only available batch by batch.

Moreover, this assumption allows us to obtain much more precise results than previous works which only obtain upper bounds on the error. We emphasize also that this assumption is somewhat standard when trying to obtain rigorous asymptotic characterization of optimization trajectories using similar techniques (e.g., reference [CPT23]). We leave relaxations of this assumption to future work, since the resulting state equations would likely be much more complicated (as mentioned by Reviewer kkyZ), and the proofs may require substantial new machinery. We will be sure to include some of these remarks as discussion points in the revised paper.


To further assess the strength of this assumption, we have attached additional simulations which compare three types of batch selection (under the same hyperparameter choices as in Figure 1a):
- Fresh batches at each iteration: this is the setting we consider in our paper and for which our theoretical results are derived
- Randomly sampled batches (with possible repeats): this corresponds to choosing $n$ samples at each iteration from a global pool of $nT$ samples, with possible repetition of data across iterations.
- Same batch: using the same $n$ samples for each iteration

We find that the theoretical results derived for Setting 1 seem to remain accurate for Setting 2, even though the batches are no longer independent (formalizing this would be an interesting technical problem for future work). However, as expected, the predictions are too optimistic for Setting 3 (which only uses $n$ total samples during training, rather than $nT$).


**(b) Motivation and connections to feature learning:** We also plan to incorporate the helpful suggestions of Reviewer yJHL to improve the motivation and contextualization of our results. Some important take-home messages we hope to highlight for our results include (i) a rigorous framework for comparing several existing (and new) algorithms from the perspective of generalization error, (ii) favorable guarantees on the test error within a constant number of iterations (which is more optimistic than the convergence results common in the literature), and (iii) provable benefits of weight-sharing in LDNNs in the presence of structured sparsity.

As a further clarification, our use of the phrase “feature learning” refers to the ability of these algorithms to learn LDNN parameters which effectively recover sparse or group-sparse signals. Note that in the case of linear models, learning “good” features amounts to learning which subset of coordinates in the input are important, so feature learning in linear diagonal networks is equivalent to learning the low-dimensional structure of the true signal (e.g., sparsity or group-sparsity structure). See also the discussion in [RMD24] for more discussion on this connection and how it was used to construct the lin-RFM algorithm (which we analyze). We will be sure to add this connection more explicitly after lines 39-40.

References:

[CPT23] Chandrasekher et al. “Sharp global convergence guarantees for iterative nonconvex optimization with random data,” The Annals of Statistics, 2023.

[RMD24] Radhakrishnan et al. “Linear Recursive Feature Machines provably recover low-rank matrices,” arXiV preprint, 2024.

---

> ### Comment · Reviewer_kkyZ · 2024-08-08
>
> Thank you for the detailed response.
>
> I have a question regarding the numerical results shown in the attached pdf.
> I thought that reusing batches is beneficial (in a somewhat different context of training say 2-layer neural nets with SGD) given a recent line of work [R1, R2, R3]. The message there is that to achieve the same error, the sample complexity is smaller (in order) if samples are reused (even twice per sample). So with the same sample size, the error of online / sample-splitting methods should be larger (cf.\ [R4, R5]).
> However, the experiment in the attached pdf suggests a seemingly opposite message since it compares sampling-splitting with reusing **a single batch**.
> This doesn't seem to be a fair comparison and indeed leads to an ``optimistic'' prediction.
> It seems more standard to compare the performance with and without sample-splitting on the same $nT$ samples.
>
> ---
>
> [R1] Lee, J. D., Oko, K., Suzuki, T., & Wu, D. (2024). Neural network learns low-dimensional polynomials with SGD near the information-theoretic limit. arXiv preprint arXiv:2406.01581.
>
> [R2] Dandi, Y., Troiani, E., Arnaboldi, L., Pesce, L., Zdeborová, L., & Krzakala, F. (2024). The benefits of reusing batches for gradient descent in two-layer networks: Breaking the curse of information and leap exponents. arXiv preprint arXiv:2402.03220.
>
> [R3] Arnaboldi, L., Dandi, Y., Krzakala, F., Pesce, L., & Stephan, L. (2024). Repetita iuvant: Data repetition allows sgd to learn high-dimensional multi-index functions. arXiv preprint arXiv:2405.15459.
>
> [R4] Damian, A., Nichani, E., Ge, R., & Lee, J. D. (2024). Smoothing the landscape boosts the signal for sgd: Optimal sample complexity for learning single index models. Advances in Neural Information Processing Systems, 36.
>
> [R5] Arous, G. B., Gheissari, R., & Jagannath, A. (2021). Online stochastic gradient descent on non-convex losses from high-dimensional inference. Journal of Machine Learning Research, 22(106), 1-51.

---

> > ### Author Response · Authors · 2024-08-09
> > **Simulations with reused samples**
> >
> > Thank you for the response and great observation – yes, in general, we would expect reusing $nT$ samples at each iteration to be better than sample-splitting (in line with the references you mentioned). We agree that this is perhaps the more standard point of comparison, and we have rerun the simulations with this additional curve. As expected, this approach does outperform the other approaches when the regularization parameter is appropriately tuned (unfortunately, we are not able to update the pdf in this response). So, to summarize, our theoretical predictions for the sample-split case
> >
> > - seem to remain accurate when randomly sampling $n$ samples from a total pool of $nT$ at each iteration
> > - are too optimistic for the scenario where $n$ samples are reused at every iteration
> > - are too pessimistic for the case where $nT$ samples are reused at every iteration.

---

> > > ### Comment · Reviewer_kkyZ · 2024-08-10
> > >
> > > Thank you very much for the prompt update.
> > > This makes perfect sense to me.

---

### Decision · Program_Chairs · 2024-09-25

**Decision:**

Accept (poster)

**Comment:**

Three reviewers evaluated the paper, and their overall assessment is positive. I concur with their evaluation and believe the paper offers a strong contribution with compelling results. Some reviewers pointed out a potential limitation related to the sample-splitting assumption. In response, the authors conducted additional simulations using various batch selection methods to address this concern. Another reviewer noted the connection between the presented work and feature learning. The authors welcomed this feedback and have committed to clarifying this connection in the paper.